# MULTI-AGENT INTERPOLATED POLICY GRADIENTS

## ABSTRACT

Policy gradient method typically suffers high variance, which is further amplified in the multi-agent setting due to the exponential growth of the joint action space. While value factorization is a popular approach for efficiently reducing the complexity of the value function, integrating it with policy gradient to reduce variance is challenging, as bias is introduced due to the limitations of factorization structure. This paper addresses the underexplored bias-variance trade-off problem by proposing a novel policy gradient method in MARL that uses a convex combination of joint Q-function and a factorized Q-function. This results in a policy gradient approach that balances stochastic and factorized deterministic policy gradients, enabling a more flexible trade-off between bias and variance. Theoretical results validate the effectiveness of our approach, showing that factorized value functions can effectively reduce variance while potentially maintaining low bias. Empirical experiments on several benchmarks demonstrate that our approach outperforms existing state-of-the-art methods in terms of efficiency and stability.

## 1 INTRODUCTION

Cooperative multi-agent reinforcement learning (MARL) has become an increasingly popular research area due to its wide range of potential applications, including robotics (Zhang et al. (2021a)), autonomous driving (Zhou et al. (2020)), inventory management (Ding et al. (2022)) and so on (Wang et al. (2021b); Yu et al. (2022b); Zhou et al. (2021)). MARL extends reinforcement learning to scenarios where multiple agents interact in a shared environment, presenting new challenges and opportunities for researchers in the field of RL. One popular learning paradigm in cooperative MARL is Centralized Training with Decentralized Execution (CTDE) (Kraemer & Banerjee (2016); Oliehoek et al. (2008)), which has shown promising results in addressing the non-stationary problem (Hernandez-Leal et al. (2017)) of MARL by allowing agents to access global information during training while still maintaining individual control during execution.

Policy gradient (Sutton et al. (1999)) is a successful method employed in CTDE for multi-agent learning, enabling agents to learn centralized value functions to optimize their individual policies while maintaining good coordination and convergence (Wang et al. (2020b); Yu et al. (2022a)). However, policy gradient methods often suffer from high variance, which is further amplified in the multi-agent setting due to the exponential growth of the joint action space, and the sub-optimal or exploratory actions from other agents. Consequently, achieving a stable and efficient policy that yields good performance in MARL poses a challenge.

Recently, value function factorization method (Rashid et al. (2020b); Son et al. (2019); Sunehag et al. (2017); Wang et al. (2020a)) has emerged as a popular approach in MARL under CTDE for efficiently learning the joint value function. By decomposing the joint value function into individual utilities of individual state-action spaces, the value factorization method reduces the complexity of the factorized function, thereby improving learning efficiency. However, this reduction in complexity comes at the cost of introducing bias due to the limited function class. While value function factorization has demonstrated success in value-based methods, its integration with policy gradient methods in MARL remains relatively unexplored, and it may not fully leverage its advantages. Several policy-based approaches (Peng et al. (2021); Wang et al. (2020b); Zhang et al. (2021b)) directly apply value factorization to learn the value function, reducing the variance of the policy gradient but also introducing some bias. The degree of bias and variance is determined by the factorization structures used, and it is difficult to achieve the optimal trade-off between them since it requires new structures. This inflexibility can limit the performance of the policy-based approach and make it challenging to

achieve good results. Furthermore, while the theoretical properties of value factorization in policy gradient have been partially explored (Wang et al. (2020b)), further investigation is necessary to advance our understanding in this field.

In this paper, we demonstrate that bias-variance trade-offs can be achieved by employing a convex combination of joint Q-function and a factorized Q-function, resulting in a Multi-Agent Interpolated Policy Gradient (MAIPG). By integrating an unbiased stochastic policy gradient with a low-variance factorized deterministic policy gradient, our approach offers a more flexible trade-off between bias and variance. Theoretical results validate the effectiveness of bias-variance trade-offs and show that the factorized value functions can effectively reduce variance while potentially maintaining low bias. To evaluate the performance of our approach, we conducted empirical experiments on several benchmark scenarios. The results show that our proposed method achieves superior bias-variance trade-offs and outperforms existing state-of-the-art methods in terms of efficiency and stability.

## 2 BACKGROUND

### 2.1 PRELIMINARIES

We employ the Multi-agent Markov Decision Process (MMDP) (Boutilier (1996)) as the framework, which is a special case that rules out the concerns of partial observability of Decentralized Partially Observable Markov Decision Process (Dec-POMDP) (Oliehoek & Amato (2016)) in modeling cooperative multi-agent tasks. It's worth noting that our method is applicable for Dec-POMDP. The choice of MMDP is to simplify the theoretical analysis. The MMDP is defined by a tuple $\langle \mathcal{N}, \mathcal{S}, \mathcal{A}, r, \mathcal{P}, \gamma \rangle$, where $\mathcal{N}$ represents the set of agents, $\mathcal{S}$ is the set of states, $\mathcal{A}$ is the set of actions, $r$ is the reward function, $\mathcal{P}$ is the transition probability function, and $\gamma$ is the discount factor. At each time step, each agent $i \in \mathcal{N}$ observes the current state $s \in \mathcal{S}$ and then selects an action $a_i \in \mathcal{A}$ based on its policy $\pi_i(a_i|s) : \mathcal{S} \to \Delta(\mathcal{A})$ (the probability simplex over $\mathcal{A}$). The actions of all agents form a joint action $a \sim \pi(a|s)$, where $\pi(\cdot|s)$ represents the joint policy formed by $\pi_i$. The next state resulted by current state and joint action is determined by the transition probability function $\mathcal{P}(s'|s,a)$, and a global reward $r$ is shared by all agents. The joint policy induces a distribution over trajectories $\tau = (s_t, a_t, r_t)_{t=0}^{\infty}$ and for all subsequent timesteps $t$, $a_t \sim \pi(\cdot|s_t)$ and $s_{t+1} \sim \mathcal{P}(\cdot|s_t, a_t)$. The objective of MARL is to find a joint policy $\pi$ that maximizes the expected discounted rewards, defined by $V^\pi(s) = \mathbb{E}\left[\sum_{t=0}^{\infty} \gamma^t r_t | \pi, s_0 = s\right]$. Consequently, the action-value function (or Q-function) is defined as $Q^\pi(s,a) = r + \gamma \mathbb{E}_{s' \sim P(\cdot|s,a)}\left[V^\pi(s')\right]$. Additionally, we define deterministic policies $\mu_i$ for each agent based on $\pi_i$, i.e. $\mu_i(s) = \text{argmax}_{a_i} \pi_i(a_i|s)$. Let $\mu(s) = \langle \mu_1(s), ..., \mu_n(s) \rangle$ be the joint deterministic policy. Further, we use $\theta_i$ denote the parameters of the individual policy $\pi_{\theta_i}$ and $\mu_{\theta_i}$. Then $\theta = \langle \theta_1, ..., \theta_n \rangle$ is the parameters of the joint policy $\pi$ and $\mu$. To maintain conciseness, notations with subscripts "$i$" pertain to individual agents, while notations without subscripts encompass all agents, or they refer to single-agent notations in the context of single-agent RL.

The centralised training with decentralised execution (CTDE) paradigm is employed in our work. CTDE allows for policy training to leverage global information that may be available, as well as sharing information between agents during training. During execution, however, each agent can only access its own action-observation history, thereby implementing decentralised execution.

### 2.2 STOCHASTIC AND DETERMINISTIC POLICY GRADIENT

Policy gradient methods can be regarded to solve the optimization problem: $\max_\pi V^\pi(s_0)$. These methods (Schulman et al. (2015a); Williams (1992)) compute gradients of the objective and update policy parameters to maximize the expected return. If the policy is stochastic, the policy gradient functional form (Sutton et al. (1999)) with respect to $\theta$ is:

$$\nabla_\theta J(\theta) = \nabla_\theta V^{\pi_\theta}(s_0) = \frac{1}{1-\gamma} \mathbb{E}_{s \sim d_\rho^{\pi_\theta}} \mathbb{E}_{a \sim \pi_\theta(\cdot|s)}\left[\nabla_\theta \log \pi_\theta(a|s) Q^{\pi_\theta}(s,a)\right], \quad (1)$$

where $d_\rho^\pi := \mathbb{E}_{s_0 \sim \rho}\left[(1-\gamma)\sum_{t=0}^{\infty} Pr^\pi(s_t = s|s_0)\right]$ is the discounted state visitation distribution over initial state distribution $\rho$. This results in a basic form of stochastic policy gradient in CTDE, where two representative variants are COMA (Foerster et al. (2018)) and MAPPO (Yu et al. (2022a)):

$$\nabla_\theta J^{\text{COMA}}(\theta) = \mathbb{E}_{s \sim d_\rho^{\pi_\theta}} \mathbb{E}_{a \sim \pi_\theta(\cdot|s)}\left[\sum_{i=1}^{n} \nabla_\theta \log \pi_i(a|s) A^{\text{COMA}}(s,a)\right], \quad (2)$$

$$\nabla_\theta J^{\text{MAPPO}}(\theta) = \mathbb{E}_{s \sim d_\rho^{\pi_\theta}} \mathbb{E}_{a \sim \pi_\theta(\cdot|s)} \Big[ \sum_{i=1}^{n} \min(r_i A_i^{\text{GAE}}, \text{clip}(r_i, 1-\epsilon, 1+\epsilon) A_i^{\text{GAE}}) \Big], \qquad (3)$$

where $A^{\text{COMA}}$ is counterfactual advantage, $A^{\text{GAE}}$ is computed using GAE (Schulman et al. (2015b)). Both methods use centralized value functions to calculate the gradients of decentralized policies.

On the other hand, the objective function with a deterministic policy $\mu_\theta(s)$ according to the deterministic policy gradient theorem (Silver et al. (2014)) can be written as:

$$\nabla_\theta J^{\text{DPG}}(\theta) = \mathbb{E}_{s \sim \mathcal{D}} \big[ \nabla_\theta \mu_\theta(s) \nabla_a Q^\mu(s,a)|_{a=\mu(s)} \big], \qquad (4)$$

where $\mathcal{D}$ is a replay buffer of any behavior policy. In MARL, a representative deterministic policy gradient method is MADDPG (Lowe et al. (2017)), which learns centralized critics for each agent to calculate individual policy gradient:

$$\nabla_{\theta_i} J^{\text{MADDPG}}(\theta_i) = \mathbb{E}_{s,a \sim \mathcal{D}} \big[ \nabla_{\theta_i} \mu_{\theta_i}(s) \nabla_{a_i} Q_i^\mu(s,a)|_{a_i=\mu_i(s)} \big]. \qquad (5)$$

## 2.3 Value Factorization Methods

In cooperative MARL, value decomposition methods are used to learn a centralized but factored action-value function efficiently under the CTDE paradigm. Two representative examples of value-based methods are VDN (Sunehag et al. (2017)) and QMIX (Rashid et al. (2020b)). VDN factors $Q^\pi(s,a)$ into a sum of the per-agent utilities: $Q^{\text{VDN}}(s,a) = \sum_{i=1}^{n} Q_i(s,a_i)$. QMIX, on the other hand, uses a hypernetwork to monotonically mix each agent's utilities: $Q^{\text{QMIX}}(s,a) = f(s, Q_1(s,a_1), ..., Q_n(s,a_n))$, where $f$ represents the hypernetwork and $\frac{\partial f}{\partial Q_i} > 0$. In addition to value-based methods, some policy-based methods also utilize factored value functions in their gradient. DOP (Wang et al. (2020b)) and FACMAC (Peng et al. (2021)) are two such examples. DOP adopts a linear factorization similar to VDN to obtain a decomposed policy gradient

$$\nabla_\theta J^{\text{DOP}}(\theta) = \mathbb{E}_\pi \Big[ \sum_i k_i(s) \nabla_{\theta_i} \log \pi_i(a_i|s) Q_i(s,a_i) \Big]. \qquad (6)$$

On the other hand, FACMAC uses QMIX to learn a centralized Q-function instead of the joint Q-functions of MADDPG. It also uses a centralized gradient estimator that optimises over the entire joint action space as following:

$$\nabla_\theta J^{\text{FACMAC}}(\theta) = \mathbb{E}_{s \sim \mathcal{D}} \big[ \nabla_\theta \mu_\theta(s) \nabla_a Q^{\text{QMIX}}(s,a)|_{a=\mu(s)} \big]. \qquad (7)$$

This helps FACMAC to overcome relative overgeneralization (Wei & Luke (2016)).

## 3 Method

In this section, our focus lies on exploring the integration of value factorization and policy gradient in MARL. To achieve this, we propose an approach that combines the joint and factorized Q-functions using a convex combination. This results in an interpolation between stochastic policy gradient and factorized deterministic policy gradient, which in turn provides bias-variance trade-offs. Additionally, we present a practical algorithm that implements this concept.

## 3.1 Multi-Agent Interpolated Policy Gradient

The value function factorization method reduces the complexity of joint Q-function by limiting the function class of it, making its expressive ability crucial. A better expressive ability often results in a better fit to the Bellman equation, i.e., lower bias. However, this comes at the cost of considering a larger function class, thus potentially increasing variance. For example, QPLEX (Wang et al. (2020a)) and QTRAN (Son et al. (2019)) have better expressive ability than QMIX and VDN, but they both need to learn a function that involves the whole joint state-action space which can result in higher variance. New factorization structures may achieve a better balance between bias and variance, although constructing them is not always straightforward. As a result, our approach considers combining a joint Q-function $Q^\pi(s,a) : \mathcal{S} \times \mathcal{A}^n \to \mathcal{R}$ and a factorized Q-function $Q^\mu(s,a) : \mathcal{S} \times n\mathcal{A} \to \mathcal{R}$, with the stochastic policy $\pi$ and deterministic policy $\mu$, respectively. Although $Q^\pi(s,a)$ is unbiased,

its large joint action space encompasses actions from other agents that may come from exploration or suboptimal policies, leading to potential high variance. In contrast, $Q^\mu(s, a)$ has a factorized action space with deterministic policy, resulting in less variance but introducing bias. To achieve a more favorable bias-variance trade-off, we use a convex combination of the two terms as the new joint Q-function $Q_{jt}(s, a)$, with a weight parameter $\nu$:

$$Q_{jt}(s, a) = (1 - \nu)Q^\pi(s, a) + \nu Q^\mu(s, a). \tag{8}$$

It's worth noting that a convex combination like (8) may not work for value-based methods (see Appendix B.1 for further details). In contrast, policy-based methods directly optimize the policy and can capture the gradient indicated by the joint Q-function, making it easier to handle bias-variance trade-offs by adjusting the weight parameter of (8). The following proposition provides the policy gradient of our method, which can be regarded as a convex combination of stochastic and deterministic policy gradient.

**Proposition 1** (multi-agent interpolated policy gradient). *Given (8), the policy gradient can be written as*

$$\nabla_\theta \hat{J}(\theta) = (1 - \nu)\mathbb{E}_{s \sim d_\rho^\pi, a \sim \pi}\big[\nabla log \pi(a|s)Q^\pi(s, a)\big] + \nu \mathbb{E}_{s \sim d_\rho^\mu}\big[\nabla \mu(s)\nabla Q^\mu(s, a)|_{a=\mu(s)}\big]. \tag{9}$$

Here, we use $\hat{J}$ to distinguish it from the original objective $J$. This result is derived directly from the stochastic policy gradient theorem (Sutton et al. (1999)) and deterministic policy gradient theorem (Silver et al. (2014)) (refer to Appendix B.2). It is important to note that these theorems require that the Q-functions conform to the Bellman equation. However, in the case of factorized Q-functions, they often fail to satisfy it. In Section 4.2, we will explore the bias introduced by this deviation. Moreover, although (9) is similar to the single-agent interpolated policy gradient (Gu et al. (2017)), they are very different in both motivations and implementations. We present a detailed comparison in Appendix A.

The first term in (9) corresponds to the stochastic part of the gradient, which is based on sampling actions from the joint policy, resulting in potentially high variance and thus leading to slow convergence. The second term corresponds to the deterministic part of the gradient, which is based on the gradient of the factorized Q-function with respect to the action, evaluated at the action selected by the deterministic policy. This term has low variance, but it introduces bias consisting of the deterministic policy and factorization. Therefore, the policy gradient theorem allows us to directly trade off bias and variance by adjusting the weight $\nu$. A more detailed analysis of the bias and variance trade-off will be presented in Section 4.

### 3.2 PRACTICAL ALGORITHM

In this subsection, we will provide the implementation of our method that employs interpolated policy gradients to trade off bias and variance. To update the policy parameters, we modify (9) as following:

$$\nabla_\theta \hat{J}(\theta) = \mathbb{E}_{s \sim d_\rho^\pi}\big[(1 - \nu)\mathbb{E}_{a \sim \pi}\big[\nabla_\theta \log \pi_\theta(a|s)\hat{A}(s, a, \tau)\big] + \nu \nabla_\theta \mu(s)\nabla_{a'}\hat{Q}(s, a')|_{a'=\mu(s)}\big], \tag{10}$$

where $\hat{A}$ and $\hat{Q}$ represent the estimated values, which will be specified later. There are two differences between (9) and (10). Firstly, we use an advantage estimator instead of the joint Q-function in the first term. This is based on the superior performance of the advantage estimator compared to directly learning a joint Q-function (Papoudakis et al. (2020); Schulman et al. (2015b)). Secondly, the state distribution of the second term is merged by the state distribution $d_\pi^\mu$, as we do not have access to $d_\rho^\mu$ when the behavior policy is $\pi$. The requirement for importance sampling is eliminated due to the deterministic policy gradient. In addition, PPO-clipping is applied to the first term for two purposes: to improve training stability and performance, and to ensure the gradient is equivalent to that of MAPPO (3) when $\nu = 0$. This allows for a direct comparison between our method and MAPPO.

The overall framework of our algorithm follows the common implementation of on-policy method that is based on CTDE. We utilized a state value network $V^\varphi(s)$ with parameter $\varphi$ to estimate $V^\pi(s)$, and the factorized Q-function $Q^\mu(s, a)$ is estimated by $\hat{Q}(s, a)$ which is formed by individual action-value networks $Q^\psi = [Q_i^{\psi_i}]_{i=1}^n$ and a mixing network $M^\omega$ with parameters $\psi$ and $\omega$, respectively. The default mixing network used is QMIX. To update the networks and compute gradients, we employ a shared episode buffer that stores trajectories of all agents. Specifically, the advantage $\hat{A}$ is computed

using GAE and the $\hat{Q}$ is learned using TD($\lambda$). Similar to DDPG, we use a target Q-network, which smoothly updates its parameters to match the factorized Q-network, to reduce overestimation.

We use policy networks $\pi_\theta = [\pi_{\theta_i}]_{i=1}^n$ to represent the stochastic policy. These networks can be shared among all agents when the agents are homogeneous. The deterministic policy $\mu_i$ shares the same parameters with $\pi_i$ and reparameterization is used to realize $\mu_i(s) = \text{argmax}_{a_i} \pi(a_i|s)$. Specifically, for discrete tasks, $\mu_i$ is reparameterized using Gumbel-Softmax (Jang et al. (2016)). In the case of continuous action space, $\mu_i$ represents the mean value of Gaussian distribution $\pi_i$. It's worth noting that we only use $\mu_i$ to sample actions in (10) when calculate gradients, which means the behavior policy is $\pi$. The training process is similar to on-policy RL methods, where agents interact with the environment and collect trajectories to the shared episode buffer, and networks are updated multiple times per mini-batch. Pseudo code for the algorithm can be found in Appendix C.

## 4 THEORETICAL ANALYSIS

This section presents theoretical analysis of the proposed method. We start by demonstrating how the factorized Q-function can act as a control variate to reduce the variance of gradient estimates. Next, by deriving performance bounds, our analysis reveals that the bias could be kept small, allowing us to achieve favorable bias-variance trade-offs. Furthermore, we explore the compatible function approximation under CTDE and demonstrate its relevance to the value factorization method. Our results indicate that the factorized value function can effectively reduce variance while even preventing the introduction of bias. The proof in this section is omitted and can be found in Appendix B.

### 4.1 THE FACTORIZED Q-FUNCTION AS A CONTROL VARIATE

A control variate is a statistical technique that can improve the accuracy and efficiency of estimators by reducing their variance. The basic idea is to introduce an additional variable into the estimation process that shares a common trend with the variable of interest. To be effective, the control variate should exhibit a high correlation with the variable of interest and be easily estimable. In the context of policy-based RL, the state value function $V^\pi(s)$ is a commonly used control variate for policy gradients. Some previous works (Gu et al. (2016; 2017); Liu et al. (2017)) have explored using an action-dependent control variate $\phi(s, a)$, such as the Q-function, to further reduce the variance. However, it has been observed (Tucker et al. (2018)) that while the "true" value of the action-dependent variate can significantly reduce the variance, the "learned" variate may not outperform a state value function. This may be due to the difficulty of accurately estimating the Q-function.

In multi-agent settings, however, the utilization of value factorization methods has shown promise for identifying a suitable control variate. The factorized Q-function is highly correlated with the original Q-function and also easier to estimate, thereby potentially serving as an effective control variate. The following proposition shows that our method can be seen as using factorized Q-function as a control variate.

**Proposition 2.** *If $\pi_{\theta_i}$ is reparameterizable and can be expressed as $a_i = f_{\theta_i}(s, \xi)$, with some random noise $\xi_i$ drawn from distribution $\pi(\xi_i)$, we can derive*

$$\mathbb{E}_{\pi(a|s)}\big[\nabla_\theta \log \pi(a|s)Q(s,a)\big] = \mathbb{E}_{\pi(a,\xi|s)}\big[\nabla_\theta f_\theta(s,\xi)\nabla_a Q(s,a)\big], \tag{11}$$

*where $f_\theta(s,\xi) = (f_{\theta_1}(s,\xi_1), ..., f_{\theta_n}(s,\xi_n))^T$. We then have*

$$\nabla_\theta \hat{J}(\theta) = (1-\nu)\mathbb{E}_{d_\rho^\pi,\pi}\big[\nabla \log \pi(a|s)\big(\hat{A}(s,a,\tau) - \hat{Q}(s,a)\big)\big] + \mathbb{E}_{d_\rho^\pi}\big[\nabla\mu\nabla\hat{Q}(s,a)|_{a=\mu(s)}\big]. \tag{12}$$

This theorem and proof is similar to Liu et al. (2017) which applies Stein control variate (Stein (1986)) to policy gradient. Here we assume that $a_i \sim \pi_{\theta_i}(a|s)$ can be regarded as $a_i = f_{\theta_i}(s, \xi_i)$ with some random noise $\xi_i$, which is consistent with the practical implementation. Taking discrete tasks as an example, $\xi_i$ could be sampled from a Gumbel distribution $\pi(\xi_i)$ and the deterministic policy is denoted as $\mu_{\theta_i}(s) = \mathbb{E}_{\pi(\xi_i)}[f_{\theta_i}(s, \xi_i)]$.

Using (12), the law of total variance gives the variance of the gradient:

$$(1-\nu)^2 \mathbb{E}_s\Big[\text{Var}_{a,\tau|s}\Big(\big(\hat{A}(s,a,\tau) - \hat{Q}(s,a)\big)\nabla \log \pi(a|s)\Big)\Big]$$
$$+ \text{Var}_s \mathbb{E}_{a|s}\Big[\Big((1-\nu)\mathbb{E}_{\tau|s,a}\big[\hat{A}(s,a,\tau)\big] + \nu\hat{Q}(s,a)\Big)\nabla \log \pi(a|s)\Big]. \tag{13}$$

Thus, the variance term of joint actions, which typically has high variance, is reduced by $\hat{Q}$ and a factor of $(1 - \nu)^2$. Furthermore, in Section 5.2, we demonstrate that the factorized Q-function is capable of reducing the variance $\text{Var}(A - Q)$ more effectively compared to the joint Q-function. In summary, incorporating the factorized Q-function as a control variate shows great promise in effectively reducing variance.

## 4.2 Performance Bounds for MAIPG

In this subsection, we analyze the bias of our policy gradient by comparing the original objective (1), denoted as $J(\pi)$, with the biased objective, denoted as $\hat{J}(\pi)$ induced by (9). As mentioned earlier, using a factorized Q-function may violate the Bellman equations and introduce bias. To be specific, we define the function class $\mathcal{Q} := \{Q | Q(s, a) = f(s, [Q_i]_{i=1}^n), Q_i \in \mathbb{R}^{|\mathcal{S} \times \mathcal{A}|}\}$, where $f$ represents the factorization structure. Due to the limited function class, it is usually the case $\mathcal{T}Q \notin \mathcal{Q}$, where $\mathcal{T}$ is the Bellman operator. Previous works (Wang et al. (2021a; 2020b);Rashid et al. (2020a)) simplified the update rule of the factorized Q-function as a regression problem denoted by the operator $\mathcal{T}_{\mathcal{D}}^{\mathcal{Q}}$ as follows:

$$Q^{(t+1)} \leftarrow \mathcal{T}_{\mathcal{D}}^{\mathcal{Q}} Q^{(t)} = \arg\min_{q \in \mathcal{Q}} \mathbb{E}_{(s,a) \sim D}\big(y^{(t)} - q(s, a)\big)^2.$$

Here $y^{(t)}$ is the target and $\mathcal{D}$ denotes state-action distribution depending on specific algorithm. In this paper, we set the target to be the true Q-function $Q^{\mu}(s, a)$ and $D$ to be the state-action distribution induced by policy $\pi$ according, namely

$$\hat{Q} = \arg\min_{q \in \mathcal{Q}} \mathbb{E}_{s \sim d_\rho^\pi, a \sim \pi}\big(Q^\mu(s, a) - q(s, a)\big)^2. \tag{14}$$

In the following, we start by providing a general bound that is applicable to all $\hat{Q}$.

**Proposition 3** (General bounds for MAIPG). *If* $\delta = \max_{s,a}\big|Q^\mu(s, a) - \hat{Q}(s, a)\big|$, $\epsilon = \max_s \big|\log \pi\big(\mu(s)|s\big)\big|$, *we have*

$$\big|J(\pi) - \hat{J}(\pi)\big| \leq \frac{2\sqrt{2\epsilon}}{(1 - \gamma)^2}\nu + \nu\delta.$$

This proposition shows that the bias mainly depend on how well the factorized $\hat{Q}$ can approximate $Q^\mu$ and the KL-divergence between $\pi$ and $\mu$. Furthermore, it suggests that the introduced bias is bounded and proportional to $\nu$. While the variance is proportional to $(1 - \nu)^2$, adjusting $\nu$ allows for a direct trade-off between bias and variance.

It's worth noting that $\epsilon$ is relatively small. This observation is due to the fact that $\mu(s) = argmax_a\pi(a|s)$. On the other hand, the value of $\delta$ might be significant since it maximizes across the complete state-action space. In order to obtain a tighter bound, we introduce $\hat{Q}$ as the linear function class, and the following proposition formally asserts that when considering a linear function class and an almost deterministic policy $\pi$, the bias can be exceedingly small.

**Proposition 4** (Bounds for linear function class). *Assume* $\nabla_a Q^\mu(s, a)$ *is L-Lipschitz and there exist* $\sigma$ *such that for any* $s$, $\sum_{a \notin \mathbb{D}(\mu(s), \sigma)} \pi(a|s) \leq \sigma^4$, *where* $\mathbb{D}(\mu(s), \sigma) = \{a | \|a - \mu(s)\|_2 \leq \sigma\}$. *Then*

$$\big|J(\pi) - \hat{J}(\pi)\big| \leq \frac{2\sqrt{2\epsilon}}{(1 - \gamma)^2}\nu + \nu c L \sigma^2 e^{\frac{\epsilon}{2}},$$

*where c is a constant.*

The assumption of Lipschitz smoothness of $\nabla_a Q$ is reasonable since the Q-functions tend to be smooth in most environments. The existence of $\sigma$ can be guaranteed since the term $\sum_{a \notin \mathbb{D}(\mu(s), \sigma)} \pi(a|s)$ monotonically decreases with increasing $\sigma$, while $\sigma^4$ monotonically increases. If the policy is nearly deterministic, it is possible to make both $\mathbb{D}(\mu(s), \sigma)$ and $\epsilon$ very small. Consequently, if the optimal policy is deterministic, the bias introduced to the objective would be negligible.

### 4.3 COMPATIBLE FUNCTION APPROXIMATION IN MARL

Similar to the single-agent case, the Q-functions we learned may not follow the true gradient. In this subsection, we explore the compatible function approximators of Q-function under CTDE such that the true gradient is preserved. It's worth noting that the results hold for both stochastic and deterministic policy gradient. However, to simplify notation, we present the results for the stochastic policy gradient here, and the result for deterministic policy in Appendix D. For conciseness, we consider the tabular case, where the dimension of $\theta_i$ is equal to the dimension of individual action space, denoted by $m = |\mathcal{A}|$. Thus $\theta = (\theta_1, ..., \theta_n)^T$ is an $mn \times 1$ column vector. We can then rewrite the gradient of the logarithm of the joint policy with respect to $\theta$ as:

$$\nabla_\theta \log \pi_\theta(a|s) = \sum_{i=1}^n \nabla_\theta \log \pi_{\theta_i}(a_i|s) = (\nabla_{\theta_1} \log \pi_{\theta_1}(a_1|s), ..., \nabla_{\theta_n} \log \pi_{\theta_n}(a_n|s))^T, \quad (15)$$

where the second equation follows from the independence between $\theta_i$ and $\theta_j$ for $i \neq j$. Based on this, we can state the following proposition, similar to the Compatible Function Approximation in single-agent RL (Sutton et al. (1999)):

**Proposition 5** (Compatible Function Approximation under CTDE). *A function approximator* $Q^w(s, a)$ *is compatible with a joint stochastic policy* $\pi_\theta(a|s)$, *i.e.* $\nabla_\theta J(\theta) = \mathbb{E}_{d_\rho^\pi, \pi}[\nabla_\theta \log \pi_\theta(a|s) Q^w(s, a)]$, *if*

*1.* $\nabla_w Q^w(s, a) = \nabla_\theta \log \pi_\theta(a|s) = (\nabla_{\theta_1} \log \pi_{\theta_1}(a_1|s), ..., \nabla_{\theta_n} \log \pi_{\theta_n}(a_n|s))^T$ *and*

*2.* $w$ *minimises the mean-squared error,* $MSE(\theta, w) = \mathbb{E}_{d_\rho^\pi, \pi}[(Q^\pi(s, a) - Q^w(s, a))^2]$.

For any stochastic policy $\pi_\theta(a|s)$, there always exists a compatible function approximator of the form $Q^w(s, a) = w^T \nabla_\theta \log \pi_\theta(a|s) + V(s)$ that satisfies condition 1. Here, $V(s)$ can be any differentiable baseline function that is independent of the action $a$. Furthermore, we can rewrite it as $Q^w(s, a) = \sum_i Q_i^{w_i}(s, a_i) + V(s)$ where $Q_i^{w_i}(s, a_i) = w_i^T \nabla_{\theta_i} \log \pi_{\theta_i}(a_i|s)$ and $w_i$ is an $m \times 1$ column vector of the $mn \times 1$ column vector $w = (w_1, ..., w_n)^T$. This implies that an approximated Q-function that is compatible can be factorized as the linear sum of the utility of each agent.

To satisfy condition 2, we need to find the parameters $w$ that minimise the MSE between $Q^w$ and the true Q-function, which is quite similar to (14):

$$w = \arg\min_w \mathbb{E}_{s \sim d_\rho^\pi, a \sim \pi}(Q^\pi(s, a) - Q^w(s, a))^2 \ where \ Q^w(s, a) = \sum_i Q_i^{w_i}(s, a_i) + V(s). \ (16)$$

Proposition 5 indicates that we can use factorized value function as a compatible function approximator to not only reduce variance but also preserve the true gradient. This proposition also suggests that the value function under CTDE naturally lends itself to factorization. To understand this, noticing that, for each state, the joint value function is of dimension $m^n$, whereas $mn$ parameters suffice to fully represent the individual policies. As a result, the joint value function needs to be reduced or projected into the same dimension before updating policy parameters. This fact can be viewed as prior knowledge within the CTDE paradigm, and value factorization emerges as a natural approach to leverage it. It should be noted that while linear factorization, such as VDN, could be compatible with the compatible function approximation theorem under CTDE, this compatibility may not hold true in practice. Learning a compatible value factorization using neural networks requires further investigation and discussion. Consequently, we defer this topic to future research.

## 5 EXPERIMENTS

In this section, we provide empirical results of MAIPG on matrix games and three widely adopted cooperative multi-agent benchmarks, including the Multi-agent Particle Environment (MPE) (Lowe et al. (2017)), StarCraftII Multi-agent Challenge (SMAC) (Samvelyan et al. (2019)) and Google Research Football (GRF) (Kurach et al. (2020)). Full experimental setups and details can be found in Appendix E.

### 5.1 MATRIX GAMES

In this section, we delve into an examination of the bias introduced by MAIPG and its potential impact. We randomly generate stateless matrix games with elements uniformly distributed in the

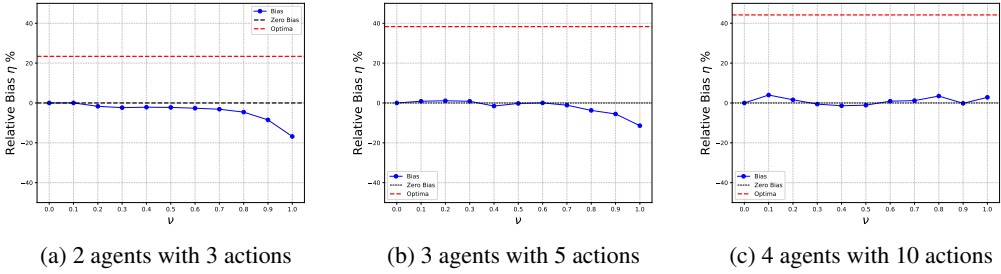

| (a) 2 agents with 3 actions | (b) 3 agents with 5 actions | (c) 4 agents with 10 actions |

Figure 1: Relative bias percentages with different $\nu$ values on randomly generated matrix games.

range $[0, 1]$, considering various numbers of agents and actions. Figure 1 displays the relative bias percentages $\eta(\nu)$ averaged over 100 randomly generated matrix games. The relative bias percentage is defined as $\eta(\nu) := (R(\nu = 0) - R(\nu))/R(\nu = 0)$, where $R(\nu)$ denotes the rewards obtained by MAIPG with the weight parameter $\nu$ after convergence averaged across these matrix games. A negative $\eta(\nu)$ indicates a biased and lower-reward situation for $R(\nu)$, while a positive $\eta(\nu)$ suggests that the results with $\nu$ are superior to the standard objective. These results reveal that, first, although $\nu = 1$ could theoretically lead to a highly biased solution, it does not consistently result in a worse solution than the standard $\nu = 0$, as illustrated in (c). Second, $\nu \in (0, 1)$ does not lead to a significant poor solution; instead, it interpolates between joint and factorized value functions (corresponding to $\nu = 0$ and $\nu = 1$, respectively). Third, the bias introduced by $\nu$ compared to the standard objective is small, especially when compared to the true optimum (red dashed line). For more comprehensive analysis please refer to Appendix F.

## 5.2 PERFORMANCE ON BENCHMARKS

**Multi-agent Particle Environment (MPE)**: Considering that this environment is relatively simple, we utilize it to evaluate the effectiveness of the factorized Q-function as a control variate, where we calculate the term $R_{\mathrm{Var}} = \mathrm{Var}(A - Q)/\mathrm{Var}A$ to provide a measure of the relative scale of variance reduction. If $R_{\mathrm{Var}} < 1$, it indicates that the variance is reduced, whereas $R_{\mathrm{Var}} \geq 1$ means that the control variate $Q$ actually increases the variance. As shown in Fig. 2, with the dashed black line representing $R_{\mathrm{Var}} = 1$, the joint Q-function exhibits significant growth in $R_{\mathrm{Var}}$ before convergence. In contrast, the factorized Q-function consistently maintains a small value of $R_{\mathrm{Var}}$, thereby effectively reducing variance. This suggests that, compared to the joint Q-function which is hard to learn and may contribute to

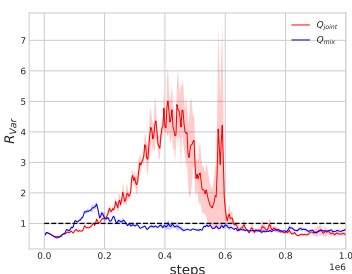

Figure 2: The relative variance $R_{\mathrm{Var}}$ on *spread* task of MPE.

increased variance, the factorized Q-function can serve as a superior control variate. We also conduct experiments on several scenarios which can be found in Appendix E.2.

**StarCraftII Multi-Agent Challenge (SMAC)**: We conducted a comparison of our method with two on-policy methods: MAPPO and HAPPO (Kuba et al. (2021a)), as well as a commonly used off-policy baseline, QMIX. The evaluation was performed on ten maps of SMAC. As shown in Fig. 8, MAIPG achieves results better than the two on-policy methods, and in most case, it performs better or comparably to the off-policy method QMIX. We also include the results of FACMAC in Appendix E.2.

**Google Research Football (GRF)**: We evaluate our algorithm with MAPPO and HAPPO in five GRF academy scenarios, including: 3v.1, counterattack easy and hard, corner, and run-pass-shoot. The QMIX is not involved due to its inferior performance reported in Yu et al. (2022a). The results, depicted in Fig. 4, demonstrate that MAIPG achieves superior performance to other methods in all settings. The experimental results on these benchmarks show that MAIPG achieves stable performance and superior efficiency compared to the state-of-the-art methods.

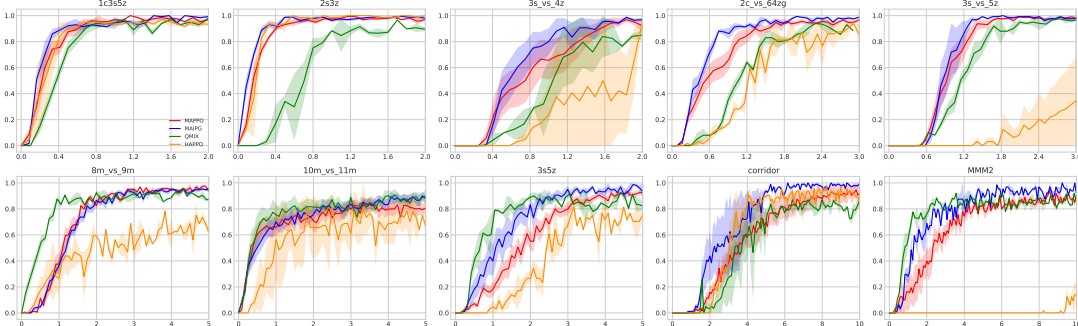

Figure 3: Mean evaluation win rate of MAIPG, MAPPO, QMIX, HAPPO in the SMAC domain, where the unit of x-axis is 1M steps and y-axis represents the test win rate.

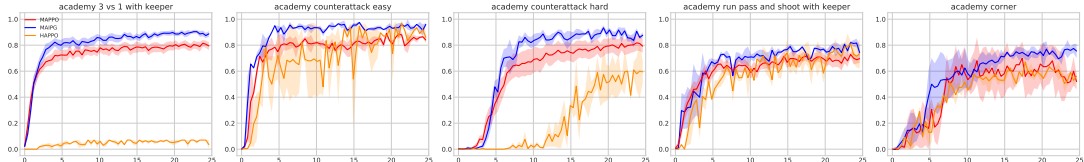

Figure 4: Mean evaluation win rate of MAIPG, MAPPO and HAPPO in the GRF domain, where the unit of x-axis is 1M steps and y-axis represents the test win rate.

## 5.3 ABLATION STUDY

In this subsection, we conduct ablation studies to investigate the impact of different weight parameters and factorization structures on gradient variance and performance. For these experiments, we select two SMAC maps as our test cases. Fig.5a illustrates the variance of three different weight parameters, namely $\nu = 0, 0.3, 0.5$, when used with QMIX's and VDN's factorization structure, respectively.

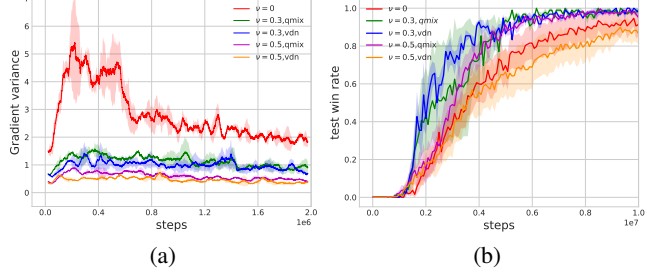

Figure 5: Ablation studies for different weight parameters and factorization structures on (a) $2s3z$ and (b) $corridor$.

We can observe that the original policy gradient ($\nu = 0$) exhibits high variance. However, as the weight parameter $\nu$ increases, the variance decreases, confirming the effectiveness of variance reduction in MAIPG. Furthermore, for a fixed $\nu$ value, the use of VDN's structure resulted in lower variance compared to QMIX, which can be attributed to the complexity of the Q-function. On the other hand, Fig. 5 shows the performance of different $\nu$ values and structures. We can observe that both QMIX and VDN structures achieved satisfactory results with appropriate values of $\nu$. It is worth noting that the performance of VDN with $\nu = 0.5$ is worse than the baseline. This may be attributed to the introduction of excessive bias caused by the weight parameter and structure. These results highlight the inherent bias-variance trade-offs associated with MAIPG.

## 6 CONSLUSION AND FUTURE WORKS

This paper focuses on mitigating the variance associated with multi-agent policy gradient methods. Our approach involves combining policy gradient and value function decomposition by employing a convex combination of joint and factorized value functions, resulting in multi-agent interpolated policy gradient (MAIPG). Theoretical analysis demonstrates that MAIPG effectively reduces variance, and the utilization of the factorized value function holds promise for minimizing bias. Experimental results further support these findings. For future work, one avenue is to achieve an unbiased compatible factorized function. Additionally, exploring the feasibility of implementing a switching control mechanism similar to MANSA to automatically adjust the weight parameter $\nu$ holds promise.

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

## A    RELATED WORKS

**Value factorization methods**. Value factorization methods such as VDN (Sunehag et al. (2017)) and QMIX (Rashid et al. (2020b)), have been developed to decompose the joint value function into individual value functions while preserving the Individual-Global-MAX (IGM) principle (Son et al. (2019)). To adhere to the IGM principle more effectively, QTRAN (Son et al. (2019)) formulates the decomposition as an optimization problem. WQMIX (Rashid et al. (2020a)) assigns higher weights to the optimal joint action values in QMIX's operator. QPLEX (Wang et al. (2020a)) introduces a duplex dueling structure to constrain the advantage functions. Both QTRAN and QPLEX can achieve the sufficient and necessary conditions for IGM. However, we do not consider QTRAN and QPLEX in this paper due to their potential for high variance.

**Multi-agent policy gradient methods**. COMA (Foerster et al. (2018)) and MADDPG (Lowe et al. (2017)) firstly introduce the paradigm of centralized critic with decentralized actors to deal with the non-stationarity issue while maintaining decentralized execution. MAAC (Iqbal & Sha (2019)) and MAPPO (Yu et al. (2022a)) build upon this paradigm by integrating it with SAC (Haarnoja et al. (2018a;b)) and PPO (Schulman et al. (2017)), respectively. DOP (Wang et al. (2020b)) is the first attempt to introduce linear value factorization to multi-agent policy gradient and formally establishes its convergence guarantee. FOP (Zhang et al. (2021b)) and FACMAC (Peng et al. (2021)) are two methods that integrate SAC with QPLEX and MADDPG with QMIX, respectively. Kuba et al. (2021b) derived the optimal baseline (OB) that achieves minimal variance in multi-agent policy gradient. While it can cooperate with our method, we opted against OB because it requires estimating the joint Q-function and is not compatible with GAE. MANSA (Mguni et al. (2023)) introduced an additional agent that uses switching controls to decide when to use independent or centralized learning. Applying this idea to our work to dynamically adjust the weight parameter $\nu$ is a promising avenue for future research. Another line is about the fully decentralized multi-agent learning (de Witt et al. (2020); Suttle et al. (2020); Yang et al. (2018); Zhang et al. (2018)), where each agent has its own reward.

**Single-agent RL**. The IPG method (Gu et al. (2017)) is particularly relevant to this paper. The basic difference between IPG and MAPIG lies in the usage of off-policy data. IPG employs off-policy updates to enhance sample efficiency, whereas MAIPG exclusively employs on-policy policy gradients to harmonize with the factorized Q-function, thereby achieving both low variance and low bias. Specifically, while IPG samples trajectories over behavior policy $\beta$, MAIPG relies on on-policy trajectories of $\pi$ as stipulated in equation (14) for theoretical underpinning. Furthermore, MAIPG incorporates a factorized Q-function, distinguishing it from IPG's use of a single off-policy critic for Q-function estimation. Along the line of control variate in RL, Q-prop (Gu et al. (2016)) firstly introduces an off-policy critic as a control variate. Additionally, (Liu et al. (2017)) introduce a more general action-dependent baseline function to improve sample efficiency. However, as pointed out in (Tucker et al. (2018)), the action-dependent baseline does not reduce variance over a state-dependent baseline in commonly tested benchmark domains.

## B    OMITTED PROOFS

### B.1    CONVEX COMBINATION FOR VALUE-BASED METHOD

In Section 3, we discovered that employing a convex combination, as shown in (8), may not effectively balance the bias and variance for value-based methods. To further illustrate this point, let's consider the Bellman operator $\mathcal{T}_D^{\mathcal{Q}}$ introduced in Section 4.2

$$\underset{\hat{Q}\in\mathcal{Q}}{\arg\min}\mathbb{E}_{(s,a)\sim D}\Big(r + \gamma\underset{a'}{max}\big((1-\nu)Q(s',a') + \nu\hat{Q}(s',a')\big) - (1-\nu)Q(s,a) - \nu\hat{Q}(s,a)\Big)^2.$$

where $Q$ represents the unfactorized value function and $\hat{Q}$ represents the factorized value function, and we use the one-step TD target. However, this formulation presents two problems. Firstly, we encounter difficulties in finding an action $a'$ that maximizes the interpolated Q-function $(1-\nu)Q+\nu\hat{Q}$. Secondly, we can observe that $Q = Q^\pi/(1-\nu)$ and $\hat{Q} = 0$ form an optimal solution. This implies that the factorized function may not be adequately learned, raising concerns about its effectiveness.

Another approach would be to learn the two Q-functions separately through their own Bellman equations. However, this renders the additional joint Q-function redundant, as the individual Q-functions are only associated with the factorized Q-function. Overall, applying an interpolated Q-function in value-based methods is not as straightforward as it is in policy-based methods. This necessitates further investigation in future research to better understand and address these challenges.

### B.2 Omitted Proofs in Section 3.1

**Proposition 1** (multi-agent interpolated policy gradient). *Given (8), the policy gradient can be written as*

$$\nabla_\theta J(\theta) = (1-\nu)E_{s\sim d_\rho^\pi, a\sim\pi}\big[\nabla log \pi(a|s)Q^\pi(s,a)\big] + \nu E_{s\sim d_\rho^\mu}\big[\nabla\mu(s)\nabla Q^\mu(s,a)|_{a=\mu(s)}\big] \quad (17)$$

*Proof.* The gradient with respect to the interpolated value function is

$$\nabla V_{jt}(s_0) = \nabla\mathbb{E}_{a\sim\pi}[Q_{jt}(s_0,a)]$$
$$= (1-\nu)\nabla\mathbb{E}_{a\sim\pi}\big[Q^\pi(s_0,a)\big] + \nu\nabla Q^\mu(s_0, \mu(s_0)).$$

According to the stochastic policy gradient theorem and deterministic policy gradient theorem, we have

$$\nabla\mathbb{E}_{a\sim\pi}\big[Q^\pi(s_0,a)\big] = \frac{1}{1-\gamma}\mathbb{E}_{s\sim d_s^\pi}\mathbb{E}_{a\sim\pi}\big[\nabla\log\pi(a|s)Q^\pi(s,a)\big]$$

and

$$\nabla Q^\mu(s_0, \mu(s_0)) = \frac{1}{1-\gamma}\mathbb{E}_{s\sim d_{s_0}^\mu}\big[\nabla\mu(s)\nabla Q^\mu(s,a)|_{a=\mu(s)}\big]$$

Therefore, we get the policy gradient over some initial state distribution $\rho$

$$\nabla_\theta J(\theta) = \mathbb{E}_{s_0\sim\rho}\big[\nabla V_{jt}(s_0)\big]$$
$$= (1-\nu)\mathbb{E}_{s\sim d_\rho^\pi, a\sim\pi}\big[\nabla\log\pi(a|s)Q^\pi(s,a)\big] + \nu\mathbb{E}_{s\sim d_\rho^\mu}\big[\nabla\mu\nabla Q^\mu(s,a)|_{a=\mu(s)}\big], \quad (18)$$

where we ignored the coefficient $(1-\gamma)^{-1}$. □

### B.3 Omitted Proofs in Section 4.1

**Proposition 2.** *If $\pi_{\theta_i}$ is reparameterizable and can be expressed as $a_i = f_{\theta_i}(s, \xi)$, with some random noise $\xi_i$ drawn from distribution $\pi(\xi_i)$, we can derive*

$$\mathbb{E}_{\pi(a|s)}\big[\nabla_\theta\log\pi(a|s)Q(s,a)\big] = \mathbb{E}_{\pi(a,\xi|s)}\big[\nabla_\theta f_\theta(s,\xi)\nabla_a Q(s,a)\big] \quad (19)$$

*then we have*

$$\nabla_\theta J(\theta) = (1-\nu)\mathbb{E}_{d_\rho^\pi,\pi}\big[\nabla\log\pi(a|s)\big(\hat{A}(s,a,\tau) - \hat{Q}(s,a)\big)\big] + \mathbb{E}_{d_\rho^\pi}\big[\nabla\mu\nabla\hat{Q}(s,a)|_{a=\mu(s)}\big] \quad (20)$$

*Proof.* We follow the proof of Liu et al. (2017) that we let $a_i = f_{\theta_i}(s, \xi+i) + \xi'$, where $\xi'$ is Gaussian noise $\mathcal{N}(0, h^2)$ and later we will take $h \to 0^+$. The joint distribution of $(a_i, \xi_i)$ given $s$ can be written as

$$\pi_i(a_i, \xi_i|s) = \pi_i(a_i|\xi_i, s)\pi(\xi_i) \propto exp\Big(-\frac{1}{h^2}(a_i - f_i(s,\xi_i))^2\Big)\pi(\xi_i)$$

Then we have

$$\nabla_{\theta_i}\log\pi_i(a_i, \xi_i|s) = \frac{1}{h^2}\nabla_{\theta_i}f_{\theta_i}(s,\xi_i)(a_i - f_{\theta_i}(s,\xi_i))$$
$$= -\nabla_{\theta_i}f_{\theta_i}(s,\xi_i)\nabla_{a_i}\log\pi_i(a_i,\xi_i|s)$$

Multiplying both sides with $\phi(s, a_i)$ and taking the expectation yield

$$\mathbb{E}_{\pi_i(a_i,\xi_i|s)}\big[\nabla_{\theta_i}\log\pi_i(a_i,\xi_i|s)\phi(s,a_i)\big] = -\mathbb{E}_{\pi_i(a_i,\xi_i|s)}\big[\nabla_{\theta_i}f_i(s,\xi_i)\nabla_{a_i}\log\pi_i(a_i,\xi_i|s)\phi(s,a_i)\big]$$
$$= \mathbb{E}_{\pi_i(\xi_i)}\big[\nabla_{\theta_i}f_i(s,\xi_i)\mathbb{E}_{\pi_i(a_i|\xi_i,s)}\big[-\nabla_{a_i}\log\pi_i(a_i,\xi_i|s)\phi(s,a_i)\big]\big]$$
$$= \mathbb{E}_{\pi_i(\xi_i)}\big[\nabla_{\theta_i}f_i(s,\xi_i)\mathbb{E}_{\pi_i(a_i|\xi_i,s)}\big[\nabla_{a_i}\phi(s,a_i)\big]\big]$$
$$= \mathbb{E}_{\pi_i(a_i,\xi_i|s)}\big[\nabla_{\theta_i}f_i(s,\xi_i)\nabla_{a_i}\phi(s,a_i)\big]$$
$$(21)$$

where the third equality comes from Stein's identity (Stein (1986)):

$$\mathbb{E}_\pi\big[\nabla_a \log \pi(a|s)\phi(s,a) + \nabla_a \phi(s,a)\big] = 0, \ \ \forall s$$

On the other hand,

$$
\begin{aligned}
&\mathbb{E}_{\pi_i(a_i,\xi_i|s)}\big[\nabla_{\theta_i} \log \pi_i(a_i,\xi_i|s)\phi(s,a_i)\big] \\
&= \mathbb{E}_{\pi_i(a_i,\xi_i|s)}\big[\nabla_{\theta_i} \log \pi_i(a_i|s)\phi(s,a_i)\big] + \mathbb{E}_{\pi_i(a_i,\xi_i|s)}\big[\nabla_{\theta_i} \log \pi_i(\xi_i|s,a)\phi(s,a_i)\big] \\
&= \mathbb{E}_{\pi_i(a_i,\xi_i|s)}\big[\nabla_{\theta_i} \log \pi_i(a_i|s)\phi(s,a_i)\big] + \mathbb{E}_{\pi_i(a_i|s)}\big[\mathbb{E}_{\pi_i(\xi_i|s,a_i)}\big[\nabla_{\theta_i} \log \pi_i(\xi_i|s,a)\big]\phi(s,a_i)\big] \\
&= \mathbb{E}_{\pi_i(a_i|s)}\big[\nabla_{\theta_i} \log \pi_i(a_i|s)\phi(s,a_i)\big].
\end{aligned}
\tag{22}
$$

By combining (21) and (22) we have

$$\mathbb{E}_{\pi_i(a_i|s)}\big[\nabla_{\theta_i} \log \pi_i(a_i|s)\phi(s,a_i)\big] = \mathbb{E}_{\pi_i(a_i,\xi_i|s)}\big[\nabla_{\theta_i} f_i(s,\xi_i)\nabla_{a_i}\phi(s,a_i)\big] \tag{23}$$

Using (15), the j-th element of column vector $\mathbb{E}_{\pi(a|s)}\big[\nabla_\theta \log \pi(a|s)Q(s,a)\big]$ has the form:

$$
\begin{aligned}
\Big(\mathbb{E}_{\pi(a|s)}\big[\nabla_\theta \log \pi(a|s)Q(s,a)\big]\Big)_i &= \Big(\mathbb{E}_{\pi(a|s)}\big[\sum_i \nabla_\theta \log \pi(a_i|s)Q(s,a)\big]\Big)_j \\
&= \Big(\sum_i \mathbb{E}_{\pi(a_i|s)}\big[\nabla_\theta \log \pi(a_i|s)\mathbb{E}_{\pi(a_{-i}|s)}\big[Q(s,a)\big]\big]\Big)_j \\
&= \Big(\sum_i \mathbb{E}_{\pi_i(a_i|s)}\big[\nabla_\theta \log \pi(a_i|s)\phi(s,a_i)\big]\Big)_j \\
&= \mathbb{E}_{\pi_j(a_j|s)}\big[\nabla_{\theta_j} \log \pi_j(a_j|s)\phi(s,a_j)\big],
\end{aligned}
\tag{24}
$$

where we let $\phi(s,a_i) = \mathbb{E}_{\pi_{-i}(a_{-i}|s)}\big[Q(s,a)\big]$.

Noticing that $\nabla_a Q(s,a) = (\nabla_{a_1}Q(s,a),...,\nabla_{a_n}Q(s,a))$ and $\nabla_\theta f_\theta(s,\xi)$ is a diagonal Jacobian matrix with diagonal element $\nabla_{\theta_i} f_i(s,\xi_i)$, the j-th element of column vector $\mathbb{E}_{\pi(a,\xi|s)}\big[\nabla_\theta f(s,\xi)\nabla_a Q(s,a)\big]$ has the form:

$$
\begin{aligned}
\Big(\mathbb{E}_{\pi(a,\xi|s)}\big[\nabla_\theta f(s,\xi)\nabla_a Q(s,a)\big]\Big)_j &= \mathbb{E}_{\pi(a,\xi|s)}\big[\nabla_{\theta_j} f_j(s,\xi_j)\nabla_{a_j}Q(s,a)\big] \\
&= \mathbb{E}_{\pi(a_i,\xi_i|s)}\big[\nabla_\theta f_i(s,\xi_i)\nabla_{a_i}\mathbb{E}_{\pi_{-i}(a_{-i}|s)}\big[Q(s,a)\big]\big] \\
&= \mathbb{E}_{\pi_j(a_j,\xi_j|s)}\big[\nabla_{\theta_j} f_j(s,\xi_j)\nabla_{a_j}\phi(s,a_j)\big]
\end{aligned}
\tag{25}
$$

Combining (23), (24) and (25), we get (19). Therefore, we can rewrite (10) as following

$$\nabla_\theta J(\theta) = (1-\nu)\mathbb{E}_{s\sim d_\rho^\pi,a\sim\pi}\big[\nabla \log \pi(a|s)\big(Q^\pi(s,a) - \hat{Q}(s,a)\big)\big] + \mathbb{E}_{s\sim d_\rho^\pi}\big[\nabla\mu\nabla\hat{Q}(s,a)|_{a=\mu(s)}\big]$$

$$\square$$

### B.4 OMITTED PROOFS IN SECTION 4.2

**Lemma 1.** *For two policies $\pi$ and $\pi'$ we have that*

$$|V^\pi(\rho) - V^{\pi'}(\rho)| \leq \frac{2}{(1-\gamma)^2}\mathbb{E}_{s\sim d_\rho^\pi}[D_{TV}(\pi,\pi')] \tag{26}$$

The proof of Lemma 1 can be found in Xu et al. (2020), where we assume that the reward $|r| \in [0,1]$ for convenience.

**Proposition 3** (Genral bounds for MAIPG). *If $\delta = \max_{s,a}\big|Q^\mu(s,a) - \hat{Q}(s,a)\big|$, $\epsilon = \max_s\big|\log \pi\big(\mu(s)|s\big)\big|$, we have*

$$\big|J(\pi) - \hat{J}(\pi)\big| \leq \frac{2\sqrt{2\epsilon}}{(1-\gamma)^2}\nu + \nu\delta$$

*Proof.* We overload notation and write $V^\pi(\rho) = \mathbb{E}_{s_0 \sim \rho}[V^\pi(s_0)]$, then

$$
\begin{aligned}
\big|J(\pi) - \hat{J}(\pi)\big| &= \big|V^\pi(\rho) - V_{jt}(\rho)\big| \\
&= \big|\mathbb{E}_{s_0 \sim \rho, a \sim \pi}\big[Q^\pi(s_0, a)\big] - (1-\nu)\mathbb{E}_{s_0 \sim \rho, a \sim \pi}\big[Q^\pi(s_0, a)\big] - \nu\mathbb{E}_{s_0 \sim \rho}\big[\hat{Q}(s_0, \mu(s_0))\big]\big| \\
&= \nu\big|\mathbb{E}_{s_0 \sim \rho, a \sim \pi}\big[Q^\pi(s_0, a)\big] - \mathbb{E}_{s_0 \sim \rho}\big[Q^\mu(s_0, \mu(s_0))\big] + \mathbb{E}_{s_0 \sim \rho}\big[Q^\mu(s_0, \mu(s_0)) - \hat{Q}(s_0, \mu(s_0))\big]\big| \\
&\leq \nu\big|V^\pi(\rho) - V^\mu(\rho)\big| + \nu\mathbb{E}_{s_0 \sim \rho}\Big[\big|Q^\mu(s_0, \mu(s_0)) - \hat{Q}(s_0, \mu(s_0))\big|\Big].
\end{aligned}
\tag{27}
$$

Using the result of Lemma 1, we get

$$
\begin{aligned}
\big|V^\pi(\rho) - V^\mu(\rho)\big| &\leq \frac{2}{(1-\gamma)^2}\mathbb{E}_{s \sim d_\rho^\pi}[D_{TV}(\pi, \pi')] \\
&\leq \frac{2}{(1-\gamma)^2}\sqrt{2\mathbb{E}_{s \sim d_\rho^\pi}\big[D_{\mathrm{KL}}\big(\pi(\cdot|s), \mu(s)\big)\big]} \\
&= \frac{2\sqrt{2}}{(1-\gamma)^2}\sqrt{-\mathbb{E}_{s \sim d_\rho^\pi}\big[\log \pi\big(\mu(s)|s\big)\big]} \leq \frac{2\sqrt{2\epsilon}}{(1-\gamma)^2},
\end{aligned}
$$

where the second inequality follows Pinsker's inequality (Csiszár & Körner (2011)) and Jensen's inequality. Note that the equality only holds for discrete action space. We omitted the discussion of continuous actions because the conclusion is similar but the expression can be complex.

Considering that $\big|Q^\mu(s_0, \mu(s_0)) - \hat{Q}(s_0, \mu(s_0))\big| \leq \max_{s,a}\big|Q^\mu(s, a) - \hat{Q}(s, a)\big| = \delta$, we get

$$
\big|J(\pi) - \hat{J}(\pi)\big| \leq \frac{2\sqrt{2\epsilon}}{(1-\gamma)^2}\nu + \nu\delta,
$$

which completes the proof. $\qquad\square$

**Proposition 4** (Bounds for linear function class). *Assume $\nabla_a Q^\mu(s, a)$ is L-Lipschitz and there exist $\sigma$ such that for any $s$, $\sum_{a \notin \mathbb{D}(\mu(s), \sigma)} \pi(a|s) \leq \sigma^4$, where $\mathbb{D}(\mu(s), \sigma) = \{a | \|a - \mu(s)\|_2 \leq \sigma\}$. Then*

$$
\big|J(\pi) - \hat{J}(\pi)\big| \leq \frac{2\sqrt{2\epsilon}}{(1-\gamma)^2}\nu + \nu c L\sigma^2 e^{\frac{\epsilon}{2}},
$$

*where c is a constant.*

*Proof.* According to (27)

$$
\begin{aligned}
\big|J(\pi) - \hat{J}(\pi)\big| &= \big|V^\pi(\rho) - V_{jt}(\rho)\big| \\
&= \nu\big|\mathbb{E}_{s_0 \sim \rho, a \sim \pi}\big[Q^\pi(s_0, a)\big] - \mathbb{E}_{s_0 \sim \rho}\big[Q^\mu(s_0, \mu(s_0))\big] + \mathbb{E}_{s_0 \sim \rho}\big[Q^\mu(s_0, \mu(s_0)) - \hat{Q}(s_0, \mu(s_0))\big]\big| \\
&\leq \nu\big|V^\pi(\rho) - V^\mu(\rho)\big| + \big|\mathbb{E}_{s \sim \rho}\big[Q^\mu(s, \mu(s)) - \hat{Q}(s, \mu(s))\big]\big|,
\end{aligned}
$$

it suffices to prove that

$$
\big|\mathbb{E}_{s \sim \rho}\big[Q^\mu(s, \mu(s)) - \hat{Q}(s, \mu(s))\big]\big| \leq c L\sigma^2 e^{\frac{\epsilon}{2}}.
$$

We consider the Taylor expansion with Lagrange remainder of $Q^\mu(s, a)$, namely,

$$
Q^\mu(s, a) = Q^\mu(s, \mu(s)) + \nabla_a Q^\mu(s, a)|_{a=\mu(s)}(a - \mu(s)) + \frac{1}{2}\nabla_a^2 Q^\mu(s, a_\xi)\|a - \mu(s)\|_2^2.
$$

The Lipschitz continuity give that

$$
\|\nabla_a Q^\mu(s, a) - \nabla_{a'} Q^\mu(s, a')\|_2 \leq L\|a - a'\|_2.
$$

Therefore, for $\forall a, a' \in \mathbb{D}(\mu(s), \sigma)$,

$$
\|Q^\mu(s, a) - Q^\mu(s, \mu(s)) - \nabla_a Q^\mu(s, a)|_{a=\mu(s)}(a - \mu(s))\| \leq \frac{1}{2}L\sigma^2.
$$

This implies that the first order Taylor expansion can approximate $Q^\mu$ with remainder of $O(L\sigma^2)$ if $a \in \mathbb{D}$. For $a \notin \mathbb{D}$, we have

$$
\|Q^\mu(s, a) - Q^\mu(s, \mu(s)) - \nabla_a Q^\mu(s, a)|_{a=\mu(s)}(a - \mu(s))\| \leq L\sqrt{mn},
$$

where we assume that $a \in [-1, 1]^{mn}$ and $mn$ is the dimension of joint action as used in Section 4.3. In fact, we can always normalize actions into $[-1, 1]$ for any task. We can also derive the Taylor expansion of $Q^\mu$ in terms of $Q_i$:

$$Q^\mu = c_0 + \sum_i \lambda_i Q_i + \sum_{i,j} \lambda_{ij} Q_i Q_j + ...,$$

where $\lambda_i = \frac{\partial Q^\mu}{\partial Q_i}$, $\lambda_{ij} = \frac{1}{2} \frac{\partial^2 Q^\mu}{\partial Q_i \partial Q_j}$ and $c_0$ is a constant. Considering that the same order Taylor expansions have the same order of reminders. We have for $\forall a \in \mathcal{D}$,

$$\left\| Q^\mu(s,a) - c - \sum_i \lambda_i Q_i(s, a_i) \right\| \leq c_1 L \sigma^2,$$

where $c_1$ is a constant. Therefor, there exist linear function $\overline{Q}(s,a)$ such that the MSE problem 14 satisfies

$$\sum_s d_\rho^\pi(s) \sum_a \pi(a|s)\big(Q^\mu(s,a) - \overline{Q}(s,a)\big)^2 = \sum_s d_\rho^\pi(s)\big(\sum_{a \in \mathbb{D}} + \sum_{a \notin \mathbb{D}}\big)\pi(a|s)\big(Q^\mu(s,a) - \overline{Q}(s,a)\big)^2$$
$$\leq c_1^2 L^2 \sigma^4 + mn L^2 \sigma^4$$
$$\leq c_2 L^2 \sigma^4,$$

where $c_2$ is another constant. As a consequence, the minimizer $\hat{Q}$ of the MSE problem 14 will have the error less than $c_2 L \sigma^2$, namely

$$c_2 L^2 \sigma^4 \geq \sum_s d_\rho^\pi(s) \sum_a \pi(a|s)\big(Q^\mu(s,a) - \hat{Q}(s,a)\big)^2$$
$$\geq \sum_s d_\rho^\pi(s)\pi(\mu(s)|s)\big(Q^\mu(s,\mu(s)) - \hat{Q}(s,\mu(s))\big)^2$$
$$\geq \sum_s \mathbb{E}_{s_0 \sim \rho}\Big[(1-\gamma)\sum_{t=0}^\infty \mathrm{Pr}^\pi(s_t = s|s_0)\pi(\mu(s)|s)\big(Q^\mu(s,\mu(s)) - \hat{Q}(s,\mu(s))\big)^2\Big]$$
$$\geq (1-\gamma)\mathbb{E}_{s_0 \sim \rho}\Big[\pi(\mu(s_0)|s_0)\big(Q^\mu(s_0,\mu(s_0)) - \hat{Q}(s_0,\mu(s_0))\big)^2\Big]$$
$$\geq (1-\gamma)e^{-\epsilon}\mathbb{E}_{s_0 \sim \rho}\Big[\big(Q^\mu(s_0,\mu(s_0)) - \hat{Q}(s_0,\mu(s_0))\big)^2\Big]$$
$$\geq (1-\gamma)e^{-\epsilon}\Big(\mathbb{E}_{s_0 \sim \rho}\big[Q^\mu(s_0,\mu(s_0)) - \hat{Q}(s_0,\mu(s_0))\big]\Big)^2.$$

We then get $\left|\mathbb{E}_{s_0 \sim \rho}\big[Q^\mu(s_0,\mu(s_0)) - \hat{Q}(s_0,\mu(s_0))\big]\right| \leq c_3 L \sigma^2 e^{\frac{\epsilon}{2}}$ which completes the proof. $\square$

## B.5 OMITTED PROOFS IN SECTION 4.3

**Proposition 5 (Compatible Function Approximation under CTDE).** *A function approximator $Q^w(s,a)$ is compatible with a joint stochastic policy $\pi_\theta(a|s)$, i.e. $\nabla_\theta J(\theta) = \mathbb{E}_{d_\rho^\pi, \pi}\big[\nabla_\theta \log \pi_\theta(a|s) Q^w(s,a)\big]$, if*

*1. $\nabla_w Q^w(s,a) = \nabla_\theta \log \pi_\theta(a|s) = (\nabla_{\theta_1} \log \pi_{\theta_1}(a_1|s), ..., \nabla_{\theta_n} \log \pi_{\theta_n}(a_n|s))^T$ and*
*2. $w$ minimises the mean-squared error, $MSE(\theta, w) = \mathbb{E}_{d_\rho^\pi, \pi}\big[\big(Q^\pi(s,a) - Q^w(s,a)\big)^2\big]$.*

*Proof.* If $w$ minimises the $MSE$ then the gradient of it w.r.t. $w$ must be zero. We then use the fact that, by condition 1, $\nabla_w Q^w(s,a) = \nabla_\theta \log \pi_\theta(a|s)$,

$$\nabla_w MSE(\theta, w) = 0$$
$$\mathbb{E}_{d_\rho^\pi, \pi}\big[\big(Q^\pi(s,a) - Q^w(s,a)\big)\nabla_\theta \log \pi_\theta(a|s)\big] = 0$$

Then we have

$$\mathbb{E}_{d_\rho^\pi, \pi}\big[Q^w(s,a)\nabla_\theta \log \pi_\theta(a|s)\big] = \mathbb{E}_{d_\rho^\pi, \pi}\big[Q^\pi(s,a)\nabla_\theta \log \pi_\theta(a|s)\big]$$

$\square$

---

**Algorithm 1.** MAIPG with recurrent neural network

---

1: Initialize policy networks $\pi_\theta$, state-value networks $V^\varphi$, action-value networks $Q^\psi = [Q_i^{\psi_i}]_{i=1}^n$ and a mixing network $M^\omega$
2: Initialize target networks: $\psi' = \psi$, $\omega' = \omega$
3: **while** $step \leq step\_max$ **do**
4:     set data buffer $D = \{\}$
5:     **for** $j = 1$ to $batch\_size$ **do**
6:        $\tau = []$ empty list
7:        initialize hidden states for $\pi_\theta$, $V^\varphi$ and $Q^\psi$
8:        Generate a trajectory and store it in $\tau$
9:        Compute advantage estimate $\hat{A}$ via GAE on $\tau$
10:      Compute target value $\hat{V}$ on $\tau$
11:      Compute target value $\hat{M}$ via TD($\lambda$) on $\tau$
12:      Split trajectory $\tau$ into chunks of length $L$
13:      **for** $l = 1$ to $T//L$ **do**
14:         $D = D \bigcup (\tau[l:l+T], \hat{A}[l:l+T], \hat{V}[l:l+T], \hat{M}[l:l+T])$
15:      **end for**
16:    **end for**
17:    **for** mini-batch $k = 1, ..., K$ **do**
18:      $b \leftarrow$ random mini-batch from $D$ with all agent data
19:      **for** each data chunk $c$ in the mini-batch $b$ **do**
20:         update RNN hidden states for $\pi, V$ and $Q$ from first hidden state in data chunk
21:      **end for**
22:    **end for**
23:    Adam updates $\theta, \varphi, \psi$ and $\omega$ with mini-batch $b$ using the computed target values and the gradients described in 10
24:    **if** $step \bmod d = 0$ **then**
25:      Update target networks: $\psi' = \alpha\psi + (1-\alpha)\psi'$, $\omega' = \alpha\omega + (1-\alpha)\omega'$
26:    **end if**
27: **end while**

---

## C   Algorithm

In this section, we present the pseudo code of our algorithms, as shown in Algorithm 1.

## D   Compatible Function Approximation for Deterministic Policy

The compatible function approximation for deterministic policy is similar to the proposition 5 for stochastic policy.

**Proposition 6** (Compatible Function Approximation for deterministic policy). *A function approximator $Q^w(s,a)$ is compatible with a joint deterministic policy $\mu_\theta(s)$, i.e. $\nabla_\theta J(\theta) = \mathbb{E}\big[\nabla_\theta \mu_\theta(s) \nabla_a Q^w(s,a)|_{a=\theta(s)}\big]$, if*

*1. $\nabla_a Q^w(s,a)|_{a=\mu_\theta(s)} = \nabla_\theta \mu_\theta(s)^T w = diag\{\nabla_{\theta_i}\mu_i(s)\}^T w$   and*
*2. $w$ minimises the $mean-squared\ error$, $MSE(\theta, w) = \mathbb{E}\big[\epsilon(s;\theta,w)^T \epsilon(s;\theta,w)\big] where$
$\epsilon(s;\theta,w) = \nabla_a Q^w(s,a)|_{a=\mu_\theta(s)} - \nabla_a Q^\mu(s,a)|_{a=\mu_\theta(s)}$*

*Proof.* The second equality of condition 1 comes from the fact that $\nabla_\theta \mu_\theta$ is a diagonal Jacobian matrix with elements $\nabla_{\theta_i}\mu_i(s)$, where we write $\mu_\theta = (\mu_{\theta_1}, ..., \mu_{\theta_1})^n$ as a column vector. If $w$ minimises the $MSE$ then the gradient of $\epsilon^2$ w.r.t. $w$ must be zero. We then use the fact that, by condition 1, $\nabla_w \epsilon(s;\theta,w) = \nabla_\theta \mu(s)$,

$$\nabla_w MSE(\theta, w) = 0$$
$$\mathbb{E}\big[\nabla_\theta \mu(s)\epsilon(s;\theta,w)\big] = 0$$

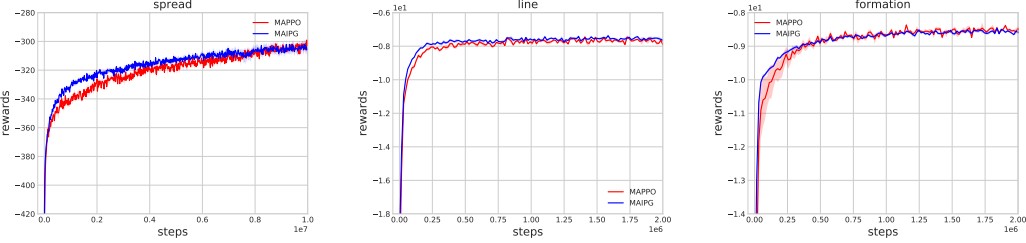

Figure 6: Mean evaluation win rate of MAIPG, MAPPO in the MPE domain.

Then we have

$$\mathbb{E}\big[\nabla_\theta \mu(s) \nabla_a Q^w(s,a)|_{a=\mu_\theta(s)}\big] = \mathbb{E}\big[\nabla_\theta \mu(s) \nabla_a Q^\mu(s,a)|_{a=\mu_\theta(s)}\big]$$

□

Similarly, we can always find compatible function approximator of the form $Q^w(s,a) = (a - \mu_\theta(s))^T \nabla_\theta \mu_\theta(s) + V(s)$ that satisfies condition 1, and it can be rewritten as $Q^w(s,a) = \sum_i Q_i(s,a_i) + V(s)$ where $Q_i(s,a_i) = (a_i - \mu_{\theta_i}(s))^T \nabla_{\theta_i} \mu_i(s)$.

# E    EXPERIMENTAL DETAILS

## E.1    IMPLEMENTATION

Our implementation is based on the MAPPO's code base. We keep the same structures and hyper-parameters, and only turn our weight parameter $\nu$ across the different tasks. Note that if we set $\nu = 0$, our algorithm is identical to MAPPO. As for the structure of the additional Q-network, we use the same architecture as the state value network. The mixing network for the QMIX's factorization structure we used, is the same as QMIX, which is a fully-connected hyper-network with two 64-dimensional hidden layers with eLU activation. The hyper-parameters in different benchmarks are basically default setting in MAPPO as presented in Table 1, Table 2, Table 3 and Table 4.

Table 1: Common hyper-parameters used across all environments.

| hyperparameters | value | hyperparameters | value | hyperparameters | value |
|---|---|---|---|---|---|
| gamma | 0.99 | optimizer | Adam | actor hidden dim | 64 |
| gae lamda | 0.95 | td lamda | 0.8 | value hidden dim | 64 |
| num mini-batch | 1 | ppo-clip | 0.2 | Q hidden dim | 64 |
| max grad norm | 10 | activation | ReLU | hidden layer | 1MLP+1GRU |

## E.2    SETUPS AND ADDITIONAL RESULTS

All the learning curves in the experiments are plotted based on several runs with different random seeds using mean and standard deviation. Specifically, MAIPG and MAPPO are averaged over at least ten seeds, and HAPPO and QMIX are averaged over three to five seeds.

**Multi-agent Particle Environment (MPE)**: The global state is formed by a concatenation of all agents' local observation since MPE does not provide it. We consider the three fully cooperative tasks: spread, line and formation (Agarwal et al. (2019)). The result is shown in Fig. 6, where we set $num\_agents = 5$ for the three tasks. The weight parameter is set to 0.05 since the reward in MPE is not normalized while the advantage function in (10) is normalized.

**StarCraftII Multi-agent Challenge**: Building upon the popular real-time strategy game StarCraft II, SMAC offers a wide range of battle scenarios that require agents to exhibit strategic thinking,

Table 2: Hyperparameters used in MPE.

| hyperparameters | value | hyperparameters | value |
|---|---|---|---|
| actor lr | 7e-4 | weight parameter | 0.05 |
| critic lr | 7e-4 | episode length | 25 |
| rollout threads | 128 | epoch | 10 |

Table 3: Hyperparameters used in SMAC.

| hyperparameters | value | hyperparameters | value |
|---|---|---|---|
| actor lr | 5e-4 | critic lr | 5e-4 |
| weight parameter | simple:0.4, hard:0.3, super hard:0.3 | rollout threads | 8 |
| epoch | simple:5, hard:10, super hard:15 | episode length | 400 |

Table 4: Hyperparameters used in GRF.

| hyperparameters | value | hyperparameters | value |
|---|---|---|---|
| actor lr | 5e-4 | episode length | 400 |
| critic lr | 5e-4 | rollout threads | 50 |
| epoch | 15 | weight parameter | 0.5 |

coordination, and adaptability. The game environment presents intricate maps, diverse unit types, and challenging objectives, all of which contribute to the complexity of the tasks. In this paper, all experiments on StarCraft II utilize the default reward and observation settings of the SMAC benchmark. We pause the training every episode and evaluate 32 episodes with individual policies to measure *win rate* of each algorithm. For each random seed, we pause the training every episode and evaluate 32 episodes with individual policies to measure *win rate* of each algorithm. Moreover, we provide additional results including FACMAC in Fig.7.

**Google Research Football**: GRF offers a set of cooperative multi-agent challenges that involve teams of agents playing against teams of bots in various football scenarios. The primary objective in these scenarios is for the agents to score goals against the opposing team. In GRF, each agent has access to complete information about the environment state through their local observations. In this paper, the dense-reward setting in GRF is employed, where all agents share a single reward. This reward is computed as the sum of individual agents' dense rewards. To evaluate the performance of the agents, the success rate is calculated based on 100 rollouts of the game.

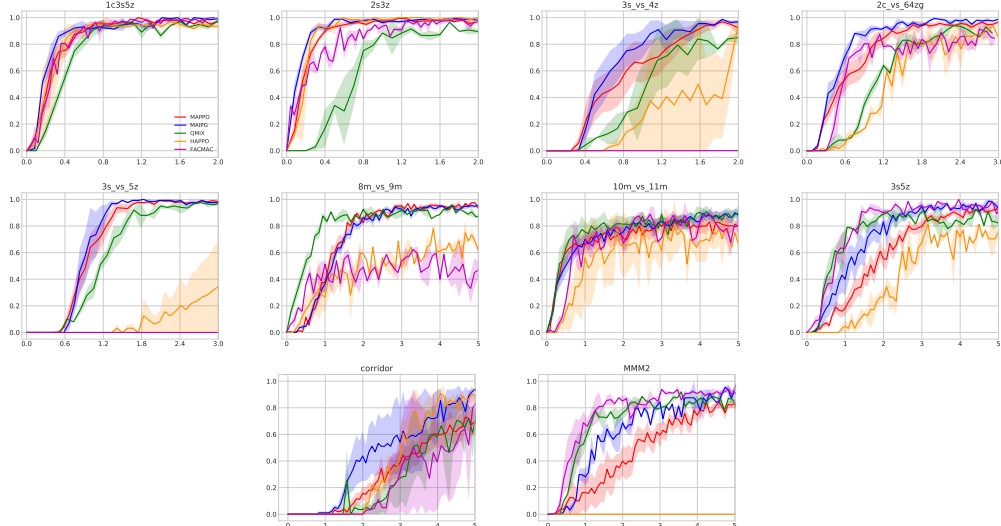

Figure 7: Mean evaluation win rate of MAIPG, MAPPO, QMIX, HAPPO and FACMAC in the SMAC domain, where the unit of x-axis is 1M steps and y-axis represents the test win rate.

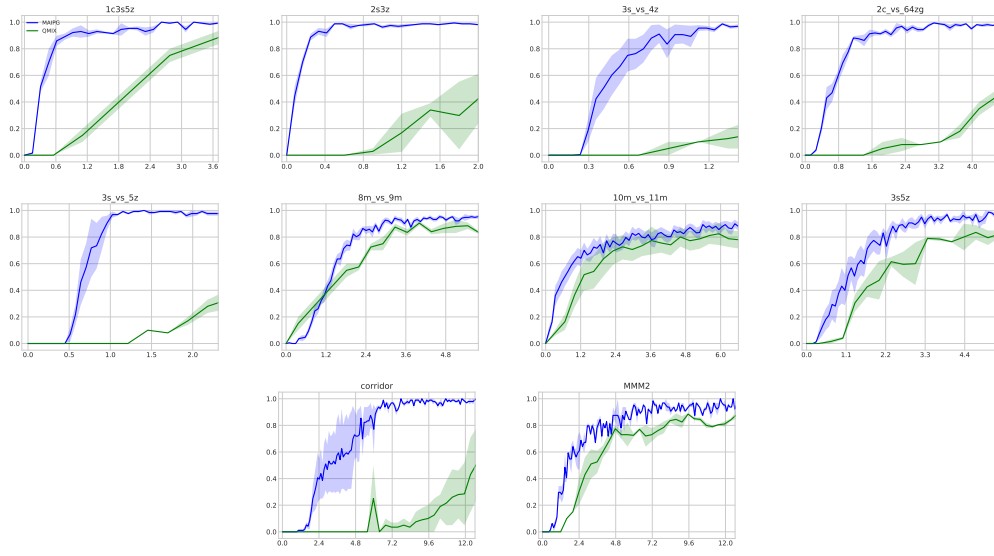

Figure 8: Mean evaluation win rate of MAIPG and QMIX in the SMAC domain, where the unit of x-axis is 1 hour on RTX3090 GPU and y-axis represents the test win rate.

.

# F  RESULTS ON MATRIX GAMES

In this section, we delve into an examination of the bias introduced by MAIPG and its potential impact. Proposition 4 establishes a bound on the bias between the standard objective and MAIPG's objective, and Proposition 5 ensures its convergence. However, concerns arise regarding whether MAIPG could lead to a suboptimal local optimum. Specifically, when $Q^\pi$ and $Q^\mu$ in Equation 8 converge to distinct maxima, there is a likelihood that a convex combination of $Q^\pi$ and $Q^\mu$ might result in a suboptimal action. To simplify, in the value-based case, we have $Q^\pi(s, a^*) \leq \max_a Q^\pi(s, a)$ and $Q^\mu(s, a^*) \leq \max_a Q^\mu(s, a)$, where $a^* = \arg\max((1 - \nu)Q^\pi(s, a) + \nu Q^\mu(s, a))$. Consequently, it is essential to demonstrate whether MAIPG's objective leads to additional suboptimal local optima.

Table 5: Cooperative matrix game. The optimal action is $(A, A)$ and sub-optimal action is $(C, C)$.

(a) Payoff of matrix game

| $a_1$ \ $a_2$ | $A$ | $B$ | $C$ |
|---|---|---|---|
| $A$ | 2 | -2 | -2 |
| $B$ | -2 | 0 | 0 |
| $C$ | -2 | 0 | 1 |

(b) Payoff of matrix game

| $a_1$ \ $a_2$ | $A$ | $B$ | $C$ |
|---|---|---|---|
| $A$ | 2 | -2 | -2 |
| $B$ | -2 | 0.5 | 0 |
| $C$ | -2 | 0 | 1 |

(c) Results of matrix game (a)

| $\nu$ \ seed | 1 | 2 | 3 | 4 | 5 | 6 | 7 | 8 | 9 | 10 |
|---|---|---|---|---|---|---|---|---|---|---|
| 0.0 | 2 | 1 | 1 | 1 | 1 | 1 | 2 | 1 | 2 | 2 |
| 0.1 | 2 | 2 | 1 | 1 | 1 | 1 | 2 | 1 | 2 | 2 |
| 0.2 | 2 | 2 | 1 | 1 | 1 | 1 | 2 | 1 | 1 | 2 |
| 0.3 | 2 | 2 | 1 | 1 | 1 | 1 | 2 | 1 | 1 | 2 |
| 0.4 | 2 | 2 | 1 | 1 | 1 | 1 | 2 | 1 | 1 | 2 |
| 0.5 | 1 | 2 | 1 | 1 | 0 | 1 | 2 | 1 | 1 | 2 |
| 0.6 | 1 | 2 | 2 | 1 | 0 | 1 | 2 | 1 | 1 | 2 |
| 0.7 | 1 | 2 | 1 | 1 | 1 | 1 | 2 | 1 | 1 | 2 |
| 0.8 | 1 | 2 | 1 | 1 | 1 | 1 | 1 | 2 | 1 | 2 |
| 0.9 | 1 | 2 | 1 | 1 | 1 | 1 | 2 | 2 | 1 | 2 |
| 1.0 | 1 | 2 | 1 | 1 | 1 | 1 | 2 | 2 | 0 | 2 |

(d) Results of matrix game (b)

| $\nu$ \ seed | 1 | 2 | 3 | 4 | 5 | 6 | 7 | 8 | 9 | 10 |
|---|---|---|---|---|---|---|---|---|---|---|
| 0.0 | 2 | 1 | 1 | 0.5 | 1 | 1 | 1 | 1 | 2 | 0.5 |
| 0.1 | 2 | 1 | 1 | 0.5 | 1 | 1 | 2 | 1 | 2 | 0.5 |
| 0.2 | 2 | 1 | 1 | 0.5 | 1 | 1 | 2 | 0.5 | 1 | 0.5 |
| 0.3 | 2 | 2 | 1 | 0.5 | 1 | 1 | 2 | 0.5 | 1 | 0.5 |
| 0.4 | 2 | 2 | 1 | 0.5 | 0.5 | 1 | 2 | 0.5 | 1 | 0.5 |
| 0.5 | 1 | 2 | 1 | 0.5 | 0.5 | 1 | 2 | 0.5 | 1 | 0.5 |
| 0.6 | 1 | 2 | 1 | 0.5 | 0.5 | 1 | 2 | 0.5 | 1 | 0.5 |
| 0.7 | 1 | 2 | 1 | 0.5 | 0.5 | 1 | 2 | 0.5 | 1 | 2 |
| 0.8 | 1 | 2 | 1 | 0.5 | 0.5 | 1 | 2 | 0.5 | 1 | 2 |
| 0.9 | 1 | 2 | 1 | 0.5 | 0.5 | 1 | 2 | 0.5 | 0.5 | 2 |
| 1.0 | 1 | 2 | 1 | 0.5 | 1 | 1 | 2 | 0.5 | 0.5 | 2 |

It is crucial to note that we cannot conclusively assert the superiority of policies induced solely by the standard objective (corresponding to $Q^\pi$) or solely by the factorized objective (corresponding to $Q^\mu$). This is due to the fact that, in comparison to the unbiased $Q^\pi$, $Q^\mu$ inherently yields biased solutions while achieving commendable results, as demonstrated in the previous works like QMIX and FACMAC. Therefore, our primary objective is to investigate whether selecting from the range $(0, 1)$ results in inferior solutions compared to the extremes of $\nu = 0$ or $\nu = 1$. For simplicity, we will refer to $Q^\pi$ in the following, even though our method exclusively utilizes $V^\pi$.

Figure 1 displays the relative bias percentages $\eta(\nu)$ averaged over 100 randomly generated stateless matrix games with elements evenly distributed in the range $[0, 1]$. The relative bias percentage is

defined as $\eta(\nu) := (R(\nu = 0) - R(\nu))/R(\nu = 0)$, where $R(\nu)$ denotes the average rewards obtained by MAIPG with the weight parameter $\nu$ after convergence in matrix games. A negative $\eta(\nu)$ indicates a biased and lower-reward situation for $R(\nu)$, while a positive $\eta(\nu)$ suggests that the results with $\nu$ are superior to the standard objective. These results reveal that, first, although $\nu = 1$ could theoretically lead to a highly biased solution, it does not consistently result in a worse solution than the standard $\nu = 0$, as illustrated in (c). Second, $\nu \in (0, 1)$ does not lead to a significant poor solution; instead, it interpolates between joint and factorized value functions (corresponding to $\nu = 0$ and $\nu = 1$, respectively). Third, the bias introduced by $\nu$ compared to the standard objective is small, especially when compared to the true optimum (red dashed line).

To further elucidate the convergence outcomes of MAIPG with varying values of $\nu$, we selected two specific matrix games labeled as (a) and (b) in Table 5. In these games, the optimal action is $(A, A)$, while the suboptimal action is $(C, C)$. Due to the non-monotonic nature of these matrix games, value factorization methods like VDN and QMIX are known to encounter challenges, often becoming stuck in the suboptimal action $(C, C)$. If the standard objective could converge to $(A, A)$, there might exist specific values of $\nu$ such that the convex combination of the two objectives would lead to the undesirable action $(B, B)$.

The outcomes of these experiments are detailed in Table 5:(c) and (d). The experiments encompassed a range of $\nu$ values, from 0 to 1, with 10 different seeds for each $\nu$. The resulting rewards after convergence were recorded, revealing several noteworthy observations. Firstly, when $\nu = 0$ or $\nu = 1$, indicating the exclusive use of either $Q^\pi$ or $Q^\mu$, suboptimal maxima or even poor maxima (Table 5(d)) are likely to be obtained in both situations. Secondly, the impact of random seeds on the convergence point is more pronounced than variations in $\nu$. Thirdly, instances of poor outcomes resulting from the convex combination of $Q^\pi$ and $Q^\mu$ (i.e., $\nu \in (0, 1)$) are seldom observed. Only in instances where $seed = 5$ and $\nu = 0.5, 0.6$ in Table 5(c), or $\nu = 0.4, 0.5, 0.6, 0.7, 0.8$ in Table 5(d), is it conceivable that these outcomes may be attributed to this combination.

Based on the above experiments, two main observations emerge. Firstly, while factorized value functions may be suboptimal in some nonmonotonic matrices, policy-based methods with the standard objective are also prone to becoming ensnared in the same suboptima in such matrices. Secondly, in an on-policy context, the convergence point of factorized value functions depends on the current policy. Since $Q^\pi$ and $Q^\mu$ are updated using the distribution induced by the same policy, they tend to share the same local optima if converged.

In conclusion, our experiments highlight the robustness of MAIPG, despite its use of a biased objective. The observed bias is minimal, and the likelihood of it leading to suboptimal policies comparable to using $Q^\pi$ or $Q^\mu$ alone. Crucially, the on-policy implementation plays a key role, ensuring the biased objective does not compromise the quality of the policy. This adaptability, coupled with the small bias and reduced variance traded off by $\nu$, underscores MAIPG's efficacy.

## G   CONVERGENCE ANALYSIS

In this section, we present an asymptotic convergence proof for our interpolated policy gradient, denoted as $\nabla \hat{J}(\theta)$ in Equation 9. The proof is a simplified version of Theorem 4.2 from Zhang et al. (2020). Let $\{\theta_k\}_{k \geq 0}$ be the sequence of parameters given by the update rule:

$$\theta_{k+1} \rightarrow \theta_k + \alpha_k \nabla \hat{J}(\theta_k), \tag{28}$$

where $\alpha_k$ is the stepsize sequence satisfies the Robbins-Monro condition:

$$\sum_{k=0}^{\infty} \alpha_k = \infty, \quad \sum_{k=0}^{\infty} \alpha_k^2 = 0$$

**Lemma 2.** *The interpolated policy gradient $\nabla \hat{J}(\theta)$ is Lipschitz continuous with some constant $L > 0$.*

*Proof.* The Lemma 3.2 of Zhang et al. (2020) gives that the standard policy gradient $\nabla J(\theta)$ is Lipschitz continuous with

$$L_1 := \frac{U_R L_\Theta}{(1 - \gamma)^2} + \frac{(1 + \gamma) U_R B_\Theta^2}{(1 - \gamma)^3},$$

where $U_R$ is the bound of reward function, $L_\Theta$ and $B_\Theta$ are Lipschitz constant and bound of $\nabla_\theta log\pi_\theta(a|s)$, respectively. The Lemma 1 of Xiong et al. (2022) establishes that the deterministic policy gradient is Lipschitz continuous with

$$L_2 := (\frac{1}{2}L_P L_\mu^2 L_d C_d + \frac{L_\psi}{1-\gamma})(L_r + \frac{\gamma U_R L_P}{1-\gamma}) + \frac{L_\mu}{1-\gamma}(L_Q L_\mu + \frac{\gamma}{2}L_P^2 U_R L_\mu C_d + \frac{\gamma L_P L_r L_\mu}{1-\gamma}),$$

where $L_P$, $L_\mu$, $L_d$, $L_\psi$, $L_r$ and $L_Q$ are Lipschitz constant of transition function $P(s'|s,a)$, deterministic policy $\mu_\theta(s)$, state visitation distribution $d^\mu(s)$, $\nabla_\theta \mu_\theta(s)$, reward function $r(s,a)$, gradient of Q function $\nabla_a Q^\mu(s,a)$, respectively. Note that all of these continuities are standard or necessary mild regularity requirements.

Now, we have

$$\|\nabla \hat{J}(\theta_1) - \nabla \hat{J}(\theta_2)\| \leq (1-\nu)L_1\|\theta_1 - \theta_2\| + \nu L_2\|\theta_1 - \theta_2\|.$$

By defining $L := (1-\nu)L_1 + \nu L_2$, we complete the proof. $\qquad\square$

Now, we define the auxiliary random variable $W_k = \hat{J}(\theta_k) - L\hat{l}^2 \sum_{j=k}^\infty \alpha_k^2$, where $\hat{l}$ is the upper bound of $\|\nabla \hat{J}(\theta_k)\|$. We have the following lemma.

**Lemma 3.** *The objective function sequence defined by (28) satisfies the following stochastic ascent property:*

$$J(\theta_{k+1}) \geq J(\theta_k) + (\theta_{k+1} - \theta_k)^T \nabla \hat{J}(\theta_k) - L\alpha_k^2 \hat{l}^2$$

*Moreover, the sequence $\{W_k\}$ is a saticifies*

$$W_{k+1} \geq W_k + \alpha_k \|\nabla \hat{J}(\theta_k)\|^2 \qquad (29)$$

*Proof.* Consider the first-order Taylor expansion of $\hat{J}(\theta_{k+1})$ at $\theta_k$. Then there exists some $\tilde{\theta}_k = \lambda\theta_k + (1-\lambda)\theta_{k+1}$ for some $\lambda \in [0,1]$ such that $W_{k+1}$ can be written as

$$W_{k+1} = \hat{J}(\theta_k) + (\theta_{k+1} - \theta_k)^T \nabla \hat{J}(\tilde{\theta}_k) - L\hat{l}^2 \sum_{j=k+1}^\infty \alpha_k^2$$

$$= \hat{J}(\theta_k) + (\theta_{k+1} - \theta_k)^T \nabla \hat{J}(\theta_k) + (\theta_{k+1} - \theta_k)^T [\nabla \hat{J}(\tilde{\theta}_k) - \nabla \hat{J}(\theta_k)] - L\hat{l}^2 \sum_{j=k+1}^\infty \alpha_k^2$$

$$\geq \hat{J}(\theta_k) + (\theta_{k+1} - \theta_k)^T \nabla \hat{J}(\theta_k) - L\|\theta_{k+1} - \theta_k\|^2 - L\hat{l}^2 \sum_{j=k+1}^\infty \alpha_k^2$$

where the inequality follows from applying Lipschitz continuity of the gradient. By definition of $W_{k+1}$, we have

$$\hat{J}(\theta_{k+1}) \geq \hat{J}(\theta_k) + (\theta_{k+1} - \theta_k)^T \nabla \hat{J}(\theta_k) - L\alpha_k^2 \hat{l}^2,$$

which establishes the first argument of the lemma. In addition, note that $\theta_{k+1} - \theta_k = \alpha_k \nabla \hat{J}(\theta_k)$, we have

$$W_{k+1} \geq W_k + \alpha_k \|\nabla \hat{J}(\theta_k)\|^2.$$

This concludes the proof. $\qquad\square$

**Proposition 5** (**Asymptotic Convergence**). *For the sequence $\theta_k$ defined by (28), we have $\lim_{k\to\infty} \theta_k \in \theta^*$, where $\theta^*$ is the set of stationary points of $\hat{J}(\theta)$.*

*Proof.* By definition, we have the boundedness of $W_k$, i.e., $W_k < J^*$, where $J^*$ is the global maximum of $\hat{J}(\theta)$. Thus, (29) can be written as

$$J^* - W_{k+1} \leq J^* - W_k - \alpha_k \|\nabla \hat{J}(\theta_k)\|^2,$$

where $\{J^* - W_k\}$ is a nonnegative bounded sequence of random variables. Therefore, we have

$$\sum_{k=0}^{\infty} \alpha_k \|\nabla \hat{J}(\theta_k)\|^2 < \infty, \ a.s. \tag{30}$$

Note the stepsize $\{\alpha_k\}$ is non-summable. Therefore, the only way that (30) may be valid is if the following holds:

$$\liminf_{k\to\infty} \|\nabla \hat{J}(\theta_k)\|^2 = 0. \tag{31}$$

So far, it suffices to show that $\limsup_{k\to\infty} \|\nabla \hat{J}(\theta_k)\|^2 = 0$. Specifically, suppose that for some random realization, we have

$$\limsup_{k\to\infty} \|\nabla \hat{J}(\theta_k)\|^2 = \epsilon > 0.$$

Then it must hold that $\|\nabla \hat{J}(\theta_k)\| \geq 2\epsilon/3$ for infinitely many $k$. Moreover, (31) implies that $\|\nabla \hat{J}(\theta_k)\| \leq \epsilon/3$ for infinitely many $k$. We thus can define the following sets $N_1$ and $N_2$ as

$$N_1 = \{\theta_k : \|\nabla \hat{J}(\theta_k)\| \geq 2\epsilon/3\}, \quad N_2 = \{\theta_k : \|\nabla \hat{J}(\theta_k)\| \leq \epsilon/3\}.$$

Note that since $\|\nabla \hat{J}(\theta_k)\|$ is continuous by Lemma 2, both sets are closed in the Euclidean space. We define the distance between the two sets as

$$D(N_1, N_2) = \inf_{\theta^1 \in N_1} \inf_{\theta^2 \in N_2} \|\theta^1 - \theta^2\|.$$

Then $D(N_1, N_2)$ must be a positive number since the sets $N_1$ and $N_2$ are disjoint and closed. Moreover, since both $N_1$ and $N_2$ are infinite sets, there exists an index set $\mathcal{I}$ such that the subsequence $\{\theta_k\}_{k\in\mathcal{I}}$ of $\{\theta_k\}_{k\geq 0}$ crosses the two sets infinitely often. In particular, there exist two sequences of indices $\{s_i\}_{i\geq 0}$ and $\{t_i\}_{i\geq 0}$ such that

$$\{\theta_k\}_{k\in\mathcal{I}} = \{\theta_{s_i}, ..., \theta_{t_i-1}\},$$

with $\{\theta_{s_i}\}_{i\geq 0} \subseteq N_1$, $\{\theta_{t_i}\}_{i\geq 0} \subseteq N_2$ and for any indices $k = s_i + 1, ..., t_i - 1 \in \mathcal{I}$ (not including $s_i$) in between the indices $\{s_i\}$ and $\{t_i\}$, we have

$$\frac{\epsilon}{3} \leq \|\nabla \hat{J}(\theta_k)\| \leq \frac{2\epsilon}{3} \leq \|\nabla \hat{J}(\theta_{s_i})\|.$$

By the triangle inequality, we may write

$$\sum_{k\in\mathcal{I}} \|\theta_{k+1} - \theta_k\| = \sum_{i=0}^{\infty} \sum_{k=s_i}^{t_i-1} \|\theta_{k+1} - \theta_k\| \geq \sum_{i=0}^{\infty} \|\theta_{s_i} - \theta_{t_i}\| \geq \sum_{i=0}^{\infty} D(N_1, N_2) = \infty. \tag{32}$$

Moreover, (30) implies that

$$\infty > \sum_{k\in\mathcal{I}} \alpha_k \|\nabla \hat{J}(\theta_k)\|^2 \geq \sum_{k\in\mathcal{I}} \alpha_k \frac{\epsilon^2}{9}$$

using the definition of $\epsilon$. We may therefore conclude that $\sum_{k\in\mathcal{I}} \alpha_k < \infty$. Given that $\|\hat{J}(\theta_k)\|$ is bounded, we have

$$\sum_{k\in\mathcal{I}} \|\theta_{k+1} - \theta_k\| = \sum_{k\in\mathcal{I}} \alpha_k \|\nabla \hat{J}(\theta_k)\| < \infty,$$

which contradicts (32). This allows us to conclude $\limsup_{k\to\infty} \|\nabla \hat{J}(\theta_k)\|^2 = 0$ almost surely. This statement together with (31) allows us to conclude that $\lim_{k\to\infty} \|\nabla \hat{J}(\theta_k)\| = 0$ almost surely, which completes the proof. $\qquad \square$

