# OpenReview forum: "Multi-Agent Interpolated Policy Gradients"
_ICLR.cc/2024/Conference — Submitted to ICLR 2024_

### Official Review · Reviewer_4F8p · 2023-11-01

**Soundness:** 3 good
**Presentation:** 3 good
**Contribution:** 2 fair
**Rating:** 5
**Confidence:** 4

**Summary:**

The authors introduce an approach to interpolate between using a joint Q function and factorised Q function in order to find a better balance in the variance bias trade-off. The idea is supported by some theoretical results that study the bias produced by the new objective in comparison with the original MARL objective.

**Strengths:**

The paper is well written and the idea, though being intuitive, seems novel and an effective tool for tackling an important problem in MARL. The authors have put effort into studying the properties of their proposed method with the theoretical analysis offering some useful insights.

The empirical results show that the method does deliver improvements in performance.

**Weaknesses:**

Despite the fact that the idea is novel, it is hard to fully evaluate the benefits of the contribution given some relevant work has seemingly been missed by the authors; specifically, these works seem highly relevant:

[1] Kuba, Jakub Grudzien, et al. "Settling the variance of multi-agent policy gradients." Advances in Neural Information Processing Systems 34 (2021): 13458-13470.

[2] Rashid, Tabish, et al. "Weighted qmix: Expanding monotonic value function factorisation for deep multi-agent reinforcement learning." Advances in neural information processing systems 33 (2020): 10199-10210.

[3] Mguni, David Henry, et al. "MANSA: learning fast and slow in multi-agent systems." International Conference on Machine Learning. PMLR, 2023.

Though [2] tackles value-based methods I think it is still worth discussing. Similarly, the method in [3] learns the best set of states to switch from using a decentralized critic to centralized critic. I caveat this with the fact that the specific algorithms involved in [3] are value-based methods but given that it is plug & play, the method seemingly captures gradient methods.Without having included a discussion on [3], it is hard to know how much this approach could be useful since the authors' approach has a fixed weighting parameter for all states whereas the approach in [3] can be viewed as a weighting variable whose optimal value {0,1} is learned for each state.


The theoretical analysis though insightful didn't allow me to fully grasp an improvement in the performance of the proposed method with regard to variance and bias. I was expecting to see some results that indicate that for a given variance, the method achieves a reduced level of bias and similarly, for given level of bias the method would achieve a lower variance as compared to the standard objective. This has been shown nicely in the ablation study but I didn't see the corresponding analytic statements for this.

It is not clear if there are situations where this method would under-perform. For example, I can see a potential for choosing greedily over either $Q^\pi$ or $Q^\nu$ yielding locally optimal actions but doing the same over their convex combination yielding a poor action. I would like to have seen a discussion on whether this is possible and under what conditions.

Minor

On page 6 it is written that the assumption of the Lipschitz smoothness of the gradient is reasonable since "Q functions tend to be smooth in most environments". I think this is slightly problematic since in many RL environments, the reward is sparse in these environments, so even the Q function is not so smooth. Besides, the smoothness of the Q functions does not suggest smoothness of its gradient.

**Questions:**

* What are the benefits/weaknesses as compared to [3]?

* For a given variance can it be shown analytically that the method achieves a reduced level of bias and similarly, for given level of bias does the method achieve a lower variance as compared to the standard objective?

* Can the authors discuss how this method would perform in situations, if they exist, where $Q^\pi$ and $Q^\nu$ may have different maxima. A coordination game such as the stag-hunt may be one such situation.



[3] Mguni, David Henry, et al. "MANSA: learning fast and slow in multi-agent systems." International Conference on Machine Learning. PMLR, 2023.

---

> ### Author Response · Authors · 2023-11-15
> **Rebuttal by Authors**
>
> We express our sincere gratitude to Reviewer 4F8p for their invaluable insights, which have significantly enhanced the robustness of our work. In response to the specific concerns raised, we present a thorough clarification of each point.
>
> 1. **Related works**.
>
> In the revised version of our paper, we have incorporated a discussion on references [a][b][c].
>
> Reference [a] is particularly relevant to our work as it addresses variance, and we emphasize that there is no conflict between [a] and our method. It is possible to integrate an optimal baseline from [a] into our method by replacing the advantage estimator in Equation 10 with Equation 12 from [a]. However, this modification requires the estimation of joint Q-functions, and GAE is not compatible, potentially impacting the performance of PPO-based gradients.
>
> Reference [b], a milestone work on value factorization, shares similar ideas when analyzing the Bellman operator of a factorized function class, and we have included it in the related works.
>
> Reference [c] proposes a promising approach using switch control to dynamically switch from independent learning to centralized learning. We believe similar methods could be applied to dynamically adjust the weight parameter $\nu$ in our approach. However, achieving this requires further efforts in both theoretical and experimental aspects, as discussed in the related and future works section of our revised paper.
>
> [a] Kuba, Jakub Grudzien, et al. "Settling the variance of multi-agent policy gradients." Advances in Neural Information Processing Systems 34 (2021): 13458-13470.
>
> [b] Rashid, Tabish, et al. "Weighted qmix: Expanding monotonic value function factorisation for deep multi-agent reinforcement learning." Advances in neural information processing systems 33 (2020): 10199-10210.
>
> [c] Mguni, David Henry, et al. "MANSA: learning fast and slow in multi-agent systems." International Conference on Machine Learning. PMLR, 2023.
>
> 2. **Analytical Comparison of Bias and Variance**.
>
> Regarding the question on achieving reduced bias and variance, our method intentionally introduces bias to reduce variance compared to the standard unbiased objective.
>
> In Equation 10, the first term, requiring a sample from the joint action space, contributes to high variance, while the second term does not. Consequently, in contrast to the standard objective ($\nu=0$), our method reduces the variance of the high variance term by a factor of $1-\nu$, simultaneously introducing a low variance term with a coefficient of $\nu$. For a more comprehensive understanding, refer to Equation 13, where the variance of the high variance term is explicitly reduced by $(1-\nu)^2$.
>
> It's important to note that, as the standard objective is unbiased, our method does not achieve reduced bias compared to it. The term "low bias" in our context reflects that the additional bias introduced by the factorized function is comparatively modest. This acknowledgment is particularly pertinent considering that a factorized function has the potential to introduce significant bias.
>
>
> 3. **Performance with Different Maxima of $Q^\pi$ and $Q^\mu$**.
>
> While we do not directly use $Q^\pi$, we understand the concern about the potential impact on policy performance if $Q^\pi$ and $Q^\mu$ have different maxima. Although combining two value functions with different maxima could lead to a suboptimal policy, we argue that such a scenario is unlikely to occur and is rarely observed. To demonstrate this, we conducted a matrix game, the details of which are provided in Appendix D3 of the revised paper. The nature of learning Q functions and our choice of on-policy implementation contribute to mitigating the likelihood of disparate maxima between $Q^\pi$ and $Q^\mu$.

---

> ### Comment · Reviewer_4F8p · 2023-11-16
> **Re:**
>
> Thanks.
>
> 1/Related works.
>
> My main concern remains as it seems that reference [c] seems to do something similar  to solving this problem but allows the value of $\nu$ to vary at each state      which seems to be a more powerful approach. In light of this I would like to have seen some statements as to why this paper is a useful contribution given [c].
>
> 2 + 3/bias
>
> My concern here is that the method introduced here seems to lead to biased solutions (and maybe even in the asymptotic training regime) - using this method it seems we lose convergence guarantees to any useful stable point. If so, because in general the stable points/equilibra of multi-agent systems are extremely sensitive to the objective parameters we could end up converging to a point a long way from any sort of local optimum (with arbitrarily bad solutions). This seems to be a significant issue.
>
> I would like to see at the very least some analysis on the conditions when this bias would be relatively small. Alternatively, if the authors could show that the bias is small in a vast number of randomly generated games that could help.

---

> > ### Author Response · Authors · 2023-11-17
> >
> > We appreciate the prompt response from reviewer 4F8p and would like to address the comments:
> > 1.  **Related works.**
> >
> > Reference [c] addresses a different problem than the one tackled in our paper. The primary contribution of MANSA lies in establishing a plug-and-play learning framework aimed at reducing the cost of centralized learning (CL) by dynamically switching between CL and independent learning (IL) while still maintaining key convergence properties.
> > It's important to note that our focus is solely on CL. While [c] mentions issues related to the complexity of CL or structural limitations on value factorization, it opts for a switch to IL rather than addressing these challenges directly. Specifically, certain challenges arise when MANSA utilizes CL, particularly in the case of SMAC, where the percentage of CL calls in MANSA is notably high (around 70%).
> >
> > In contrast, our paper directly tackles the intricacies under CL, associated with joint value functions and the biases resulting from value factorization, providing a comprehensive analysis of the bias-variance trade-off problem. Consequently, these two methods serve distinct purposes. While dynamically adjusting $\nu$ like MANSA presents a promising avenue for future work, the existence of MANSA does not diminish the significance of our contribution.
> >
> > 2. **Bias.**
> >
> > While we acknowledge the reviewer's concern regarding the bias introduced by our method, it is essential to clarify that this bias does not negate the convergence guarantees of our objective function. We assert that the similar asymptotic convergence properties observed in standard policy gradients, as exemplified in [d]’s Theorem 4.2, are applicable to demonstrate the convergence of $\theta$ to the first-order stationary point of $\hat{J}(\theta)$ in our paper. To establish this, we only need to additionally prove the Lipschitz continuity of $\hat{J}(\theta)$. Given that the first term of $\hat{J}(\theta)$ adheres to the standard objective, and the Lipschitz continuity of the second term is addressed in [e]’s Lemma 1, we substantiate our claim.
> >
> > Addressing the concern regarding biased solutions, we conducted experiments in response to the reviewer's feedback, as detailed in figure 8. It is crucial to note that many methods utilizing value factorization, such as value-based VDN, QMIX, WQMIX, and policy-based DOP and FACMAC, inherently yield biased solutions while achieving commendable results. Specifically, FACMAC, an off-policy variant of our method with $\nu=1$, serves as a pertinent example.
> >
> > Therefore, our primary objective is to establish that selecting $\nu$ from the range (0, 1) does not result in inferior solutions compared to the extremes of $\nu=0$ or $\nu=1$. Rather than quantifying the degree of bias introduced by $\nu$, it is pertinent to highlight that even QMIX, a highly biased method in non-monotonic tasks, has proven effective. The results presented in Appendix D3 demonstrate that our method does not lead to inferior solutions compared to either joint or factorized methods. Additionally, figure 8 serves to further illustrate the trade-offs between joint and factorized value functions (corresponding to $\nu=0$ and $\nu=1$, respectively).
> >
> > [d] Zhang K, Koppel A, Zhu H, et al. Global convergence of policy gradient methods to (almost) locally optimal policies[J]. SIAM Journal on Control and Optimization, 2020, 58(6): 3586-3612.
> >
> > [e] Xiong H, Xu T, Zhao L, et al. Deterministic policy gradient: Convergence analysis[C]//Uncertainty in Artificial Intelligence. PMLR, 2022: 2159-2169.

---

> ### Comment · Reviewer_4F8p · 2023-11-17
> **Re:**
>
> Thanks. These responses have allayed my concerns.
>
> If the authors could incorporate their responses, particularly, precise technical statements for point 2 in the script, I would be happy to raise my original score.
>
> Apart from that, I believe that the empirical support for the authors' claim in point 2 would be greatly strengthened if the paper included supporting experiments across a range of (say, randomly generated) normal form games (for which standard methods converge).

---

> > ### Author Response · Authors · 2023-11-19
> >
> > Thanks.
> >
> > We have categorized the responses into two sections and incorporated them into the paper revision. To be more specific, Appendix E outlines additional experiments and analyses pertaining to bias, while Appendix F provides evidence of convergence.
> >
> > Our intention is to restructure the paper to integrate the content from these sections into the main text. However, due to constraints on pages and time, we anticipate completing this integration in the next version.
> >
> > Should you have any additional concerns upon reviewing the uploaded revision paper, please feel free to inform us.

---

### Official Review · Reviewer_ZGxu · 2023-11-07

**Soundness:** 3 good
**Presentation:** 3 good
**Contribution:** 2 fair
**Rating:** 6
**Confidence:** 4

**Summary:**

In order to explore the bias-variance trade-off of the policy gradient in MARL, this paper considers a convex combination of the joint Q-function (with coefficient $1- \nu$) and a factorized Q-function (with coefficient $\nu$) and applies the policy gradient to this new function. They then establish some bounds on the bias of this function ($\propto \nu$) and the variance of its gradient ($\propto (1 - \nu)^2$). Finally, they provide some experiments for different values of $v$ to support their results.

**Strengths:**

(1) The paper investigates the idea of using this convex combination for the multi-agent case, which, based on their claims, was only done before for the single-agent case.

(2) There is a good variety of experiments that support the arguments made in the paper and also show the algorithm is flexible in the sense that it can employ different value factorization methods.

**Weaknesses:**

(1) Although this paper is for the multi-agent case and also employs a different approach in the implementation (on-policy instead of off-policy), the idea and the bounds are too similar to that of the single-agent paper referenced in section 5. In this review's opinion, this makes the result incremental and not novel and significant enough for consideration at this venue.

(2) A number of inconsistencies with the notations that make it hard for the average reader to precisely follow the claims of the paper. For instance, the notations $\nabla J$ and $\nabla \hat{J}$ are used interchangeably whereas they are not the same (look at and compare equations (1), (10), and (12)). Also, $\hat{Q}$ is used in equation (12) despite it not being properly defined until equation (14). Plus, sometimes the notations $\hat{Q}$ and $Q^\mu$ are used interchangeably even though they are clearly different when $Q^\mu$ is not in the function class $\mathcal{Q}$.

(3) There seems to be a mistake in the proof of Proposition 4. The upper bounds $\frac{1}{2} L \sigma^2$ and $L \sqrt{mn}$ are supposed to be on the absolute value of the difference of $Q^\mu (s,a)$ and its first order estimate (initialed at $\mu(s)$); however, they are apparently used as upper bounds on $(Q^\mu (s,a) - \overline{Q}(s,a))^2$. So it seems the result should have been something like $c_1 L^2 \sigma^4 + L^2 mn \sigma^2$ instead of $c_1 L \sigma^2 + L \sqrt{mn} \sigma^2$ which would change the final bound as well.
On another note, the absolute value should be outside the expectation for the second term in the last inequality of equation (27), and that is what makes the last argument of the proof (Proposition 4) possible.

**Questions:**

None.

---

> ### Author Response · Authors · 2023-11-15
> **Rebuttal by Authors**
>
> We extend our sincere appreciation to Reviewer ZGxu for the thorough evaluation of our manuscript. We are grateful for the time and dedication you have invested in providing constructive feedback. Below, we address each of the reviewers' concerns individually.
>
> 1.**The Incremental Nature of Results**.
>
> We would like to clarify certain aspects where there might be a misunderstanding regarding the novelty and significance of our work.
>
> Firstly, our approach significantly deviates from single-agent papers referenced in Section 5. The integration of a joint Q-function and factorized function in our work is motivated by a distinct trade-off between expressive ability and learning difficulty. As detailed in Appendix A.1, our exploration of a convex combination highlighted its unsuitability for value-based methods, leading us to focus on policy-based methods.
> Although the resulted policy gradients share some similarity with IPG, our motivation is fundamentally different from any single-agent paper referenced.
>
> Secondly, our theoretical results, particularly Proposition 4, provide distinctive insights into bounding the objective function. The analysis of bias induced by a factorized Q function, discussed in Section 4.2, represents a novel contribution absent in single-agent settings.
> Additionally, in Section 4.3, we introduced the concept of compatible function approximation in multi-agent reinforcement learning, exploring its relationship with value factorization.
>
> Thirdly, the “different approach in the implementation (on-policy instead of off-policy)” is not an arbitrary choice. It aligns seamlessly with our theoretical findings, as used in Equation 14 and later on, and serves to prevent poor performance, as detailed in Appendix F. This approach is a deliberate and well-justified decision based on our theoretical framework.
>
> In summary, while there may be similarities in addressing the bias-variance trade-off, our work represents a distinct and valuable contribution to the field.
> The strong relationship between the idea and bounds to value factorizations in the multi-agent settings sets our paper apart.
> Our implementation choices are grounded in the theory and have been carefully considered to yield consistent and meaningful experimental results. We believe these aspects collectively establish the novelty and significance of our proposed method.
>
> 2. **Inconsistent Notations**.
>
> We appreciate the reviewer's observation regarding inconsistent notations, and we would like to provide further clarification. In our paper, $J$ denotes the standard unbiased objective for policy gradient, while $\hat{J}$ represents the objective of our proposed method. Additionally, $\hat{Q}$ is introduced in the third line of the second paragraph of section 3.2, serving as an estimation of $Q^\mu$. To enhance consistency, we have updated the notations in the paper. Specifically, in Equation 12, we replaced $J$ with $\hat{J}$, and in Equation 10, $Q^\mu$ has been replaced by $\hat{Q}$.
>
> 3. **Mistake in the Proof of Proposition 4**.
>
> Certainly, we acknowledge the oversight in Proposition 4. In response, we have revised both the proposition and its proof in the paper. Notably, we have refined the definition of $\sigma$ for conciseness.

---

> > ### Comment · Reviewer_ZGxu · 2023-11-22
> >
> > Thank you for the clarification and the changes made in the revision. I understand the novelty of the work better now.
> >
> > The paper is much more readable now due to fixing the notations, and the proof is also taken care of. I am happy to change my original score.

---

### Official Review · Reviewer_JyH2 · 2023-11-09

**Soundness:** 3 good
**Presentation:** 2 fair
**Contribution:** 3 good
**Rating:** 6
**Confidence:** 2

**Summary:**

The paper considers the setting of multi-agent reinforcement learning and proposes the method Multi-Agent Interpolated Policy Gradient which allows for trade-off between variance and bias. Theoretical analysis of the proposed method gives an expression for the variance of the gradient and shows the effectiveness of the method in reducing variance. An upper bound on the bias introduced by incorporating a factorized Q function is also given and it is shown how by tuning a parameter bias and variance can be balanced. Finally empirical results are presented that compare the performance of the proposed method with other baselines and also ablation studies are conducted that study the effect of various design choices.

**Strengths:**

The proposed method uses a control variate, in this case factorized Q function, to reduce the variance of the policy gradient. The idea has been explored in the setting of single agent reinforcement learning and the paper extends it to multi-agent setting. Although the idea behind the main method is not original it is still a solid one and the extension of the same to multi-agent setting is significant.

The main body of the paper is presented clearly for most part and the flow of the contents is also natural.

**Weaknesses:**

The empirical results for the proposed method are not convincing. Although in GRF domain the proposed method gives better performance as compared to baselines the same is not true for the SMAC domain. If results on another benchmark could be provided it would make the empirical results stronger.

**Questions:**

1. In Algorithm 1, you mention that you use recurrent neural networks in policy networks $\pi_\theta$, state-value networks $V^\varphi$ and action-value networks $Q^\psi$. Given that in your setting each agent observes the whole state I don't see why you need it? Also, how would the performance be affected by its absence?

2. In proof of Proposition 2, you prove equation 19 first and then use that to rewrite eq. 10. What are the intermediate steps involved in this?

3. Effect of bias on the performance of the algorithm: Does a low value of $\nu$ give a better performance in most of the scenarios? I know that the effect of $\nu$ is investigated in ablation studies but since the number of experiments there is small I was wondering if the behavior seen in ablation studies holds in general or not.

4. For the GRF domain, QMIX was not included in the baselines. Is there a reason for this?

---

> ### Author Response · Authors · 2023-11-15
> **Rebuttal by Authors**
>
> We express our gratitude to Reviewer JyH2 for recognizing the strength of our work. We now address the concerns raised by the reviewer:
>
> 1. **Concerns about the Empirical Results**.
>
> While our method outperforms other baselines, it is essential to address the QMIX's comparable performance in several SMAC maps. Notably, QMIX is an off-policy method, incurring higher computational intensity and operating at about one-third the speed of our method. We have incorporated an additional plot illustrating compute time on the x-axis for a comprehensive comparison, as shown in Figure 8 of the revision of our paper.
>
> 2. **Rationale for Utilizing Recurrent Neural Networks**.
>
> As indicated in the fourth line of the preliminary section, our approach is designed for Dec-POMDP, and the choice of full observation is made to simplify theoretical analysis. Given that SMAC is partially observable, the integration of RNNs proves beneficial for enhancing performance.
>
> 3. **Clarification on Intermediate Steps from Equation 19 to Equation 10**.
>
> We can directly define
>  $\mu(s) := E_{\pi(\xi)}[f(s,\xi)]$
>  , then
>  $E_{\pi(a,\xi|s)}$$[\nabla_\theta f_\theta(s,\xi)$$\nabla_aQ(s,a)]=
> E_{\pi(a|s)}[\nabla_\theta \mu(s)\nabla_aQ(s,a)]$.
> In other words, in the context of a Gaussian policy, where $a= f_\theta(s,\xi)=\mu_\theta(s)+\Sigma_\theta(s)^{1/2}\xi$ and $\xi\sim\mathcal{N}(0,1)$,
> we have
> $E_{\pi(\xi)}[f(s,\xi)] = \mu_\theta(s)$.
>
> 4. **Impact of $\mu$ Value on Performance Across Scenarios**.
>
> In our experiments, we observed that a $\mu$ value of 0.3 in SMAC and 0.5 in GRF yielded optimal performance, which can hardly be considered "low". However, in tasks with unnormalized reward, the choice of $\mu$ depends on the reward scale. Notably, PPO employs normalized advantage, resulting in $Q^\mu$ values in our methods that may be one order of magnitude larger than $A^\pi$, necessitating a smaller $\mu$. For instance, in MPE, we set $\mu=0.05$.
>
> 5. **Exclusion of QMIX in GRF**.
>
> Addressing the absence of QMIX in GRF, as explained in the third paragraph of section 6.1, QMIX was not included due to its reported inferior performance in the MAPPO's paper.

---

> > ### Comment · Reviewer_JyH2 · 2023-11-19
> >
> > Thank you for the clarifications.
> >
> > 3/Intermediate steps: In equation 10, there is advantage function $\hat{A}$ but when you rewrite it the advantage function is replaced with $Q^\pi$. Is this a simple replacement or is there something more to it?
> >
> > Additionally, could you expand on the motivation for compatible function approximation.

---

> > > ### Author Response · Authors · 2023-11-19
> > >
> > > Thank you for your feedback.
> > >
> > > 6. **The replacement from Q to A**.
> > >
> > > This is a straightforward substitution stemming from $E_{a\sim\pi}[\nabla\log\pi(s,a)Q^\pi(s,a)]= E_{a\sim\pi}[\nabla\log\pi(s,a)A^\pi(s,a)]$
> > >
> > > 7. **The motivation for compatible function approximation**.
> > >
> > > Given our assertion in Section 4.2 that the objective function exhibits low bias, we delve into the discussion of compatible function approximation here. This serves to further demonstrate that employing factorized functions in policy gradient methods may maintain an unbiased approach. This discussion underscores the importance of investigating value factorization within a policy-based framework.

---

### Author Response · Authors · 2023-11-21

Dear Reviewers,

We appreciate the valuable feedback from the reviewers aimed at enhancing the quality of our paper. Below is a summary of the main changes made during the revision process:

- Ensured consistency and readability by modifying certain notations and rectifying errors.
- Incorporated an additional plot depicting the compute time of MAIPG and QMIX on the x-axis for comparative analysis (Figure 8).
- Moved the related works section to Appendix A and included additional related works to provide a more comprehensive context.
- Conducted new experiments on randomly generated matrix games to illustrate the bias introduced by our proposed method (Section 5.1).
- Performed a convergence analysis of the interpolated policy gradient objective (Appendix G).

We believe that these refinements significantly enhance the substance of our original submission. We welcome any further inquiries or concerns during this discussion period and are eager to address them to ensure a comprehensive understanding of our research.

---

### Meta-Review · Area_Chair_hbEy · 2023-12-07

**Metareview:**

The reviewers felt the contribution of this paper were borderline given the presence of similar approaches in the existing literature. Additionally, while the empirical results showed an improvement over other methods, the paper would benefit from a more extensive comparison.

**Justification For Why Not Higher Score:**

A more extensive empirical comparison would be a key factor that might justify the future acceptance of this paper at an ML conference.

**Justification For Why Not Lower Score:**

N/A

---

### Decision · Program_Chairs · 2024-01-16

Reject